# Patterns of interdivision time correlations reveal hidden cell cycle factors

Fern A Hughes[1,2], Alexis R Barr[2,3]*, Philipp Thomas[1]*

[1]Department of Mathematics, Imperial College London, London, United Kingdom; [2]MRC London Institute of Medical Sciences, London, United Kingdom; [3]Institute of Clinical Sciences, Imperial College London, London, United Kingdom

**Abstract** The time taken for cells to complete a round of cell division is a stochastic process controlled, in part, by intracellular factors. These factors can be inherited across cellular generations which gives rise to, often non-intuitive, correlation patterns in cell cycle timing between cells of different family relationships on lineage trees. Here, we formulate a framework of hidden inherited factors affecting the cell cycle that unifies known cell cycle control models and reveals three distinct interdivision time correlation patterns: aperiodic, alternator, and oscillator. We use Bayesian inference with single-cell datasets of cell division in bacteria, mammalian and cancer cells, to identify the inheritance motifs that underlie these datasets. From our inference, we find that interdivision time correlation patterns do not identify a single cell cycle model but generally admit a broad posterior distribution of possible mechanisms. Despite this unidentifiability, we observe that the inferred patterns reveal interpretable inheritance dynamics and hidden rhythmicity of cell cycle factors. This reveals that cell cycle factors are commonly driven by circadian rhythms, but their period may differ in cancer. Our quantitative analysis thus reveals that correlation patterns are an emergent phenomenon that impact cell proliferation and these patterns may be altered in disease.

## Editor's evaluation

This work makes an important contribution to the study of the cell cycle and inferring mechanisms by studying correlations in division timing between single cells. By treating the problem in a general way and computing over lineage trees, the authors can infer timescales in the underlying mechanism. The method is validated on data sets from bacterial and mammalian cells and can suggest when additional measurements are needed to distinguish competing models.

**\*For correspondence:**
a.barr@lms.mrc.ac.uk (ARB);
p.thomas@imperial.ac.uk (PT)

**Competing interest:** The authors declare that no competing interests exist.

## Introduction

Cell proliferation, the process of repeated rounds of DNA replication and cell division, is driven by multiple cell extrinsic and intrinsic factors (*Matson and Cook, 2017*; *Darzynkiewicz et al., 1982*). Stochasticity in any or all of these factors therefore influences the time taken for a cell to divide, generating heterogeneity in cell cycle length, even in genetically identical populations. For example, stochastic gene expression (*Elowitz et al., 2002*) can lead to heterogeneity in cell cycle length (*Kiviet et al., 2014*; *Ghusinga et al., 2016*; *Thomas et al., 2018*) as these fluctuations can be propagated by concerted cellular cues (*Co et al., 2017*). These cues can exhibit reproducible stochastic patterns that are important in development, homeostasis and ultimately, for cell survival (*Raser and O'Shea, 2005*).

Single-cell technologies illuminate a world of cellular variation by replacing bulk-average information with single-cell distributions. A key challenge is to exploit cell-to-cell variability to identify the mechanisms of cellular regulation and responses (*Raser and O'Shea, 2005*; *Martins and Locke, 2015*). Time-lapse microscopy allows us to resolve cell dynamics such as division timing, growth and

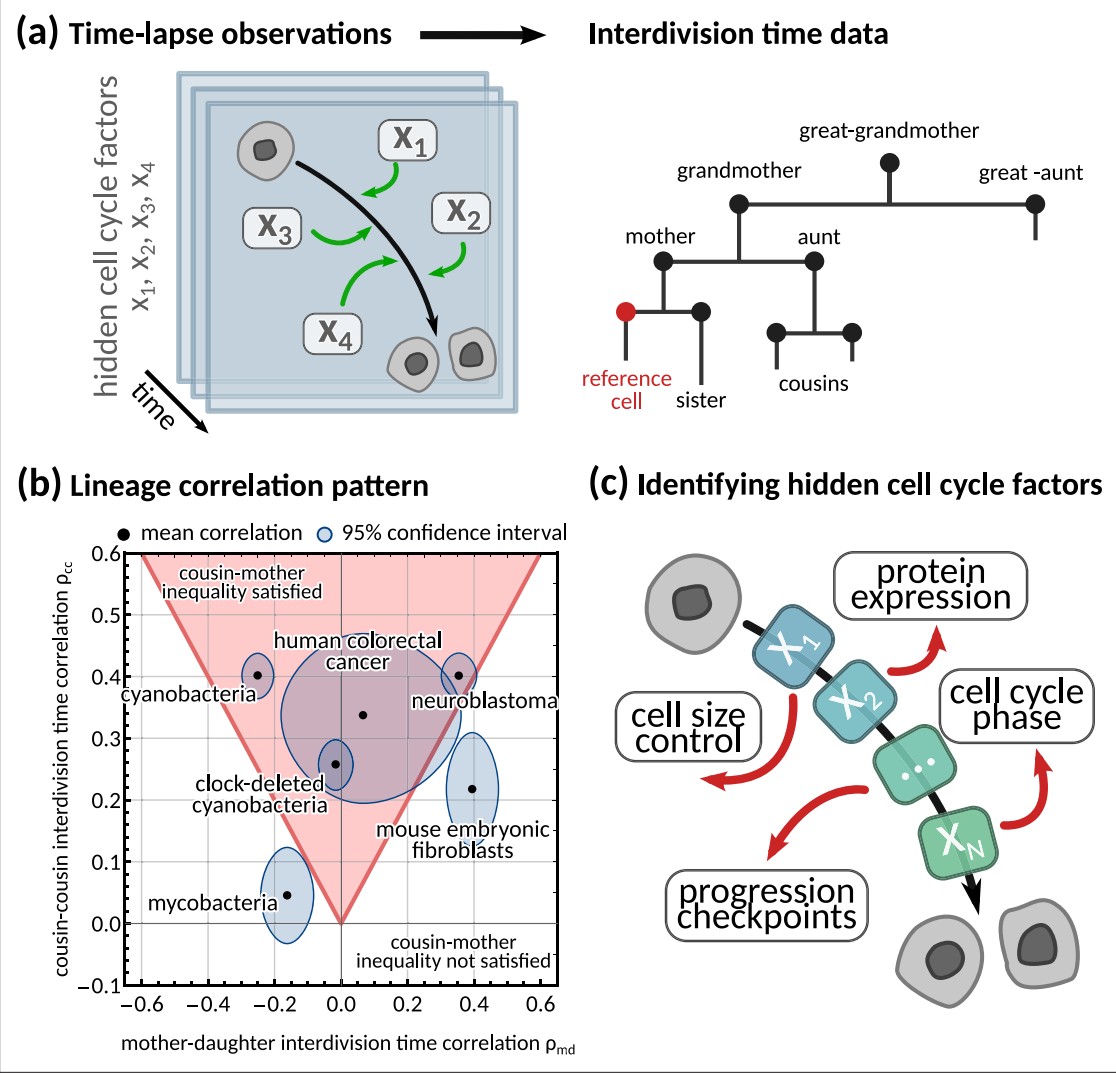

**Figure 1.** Using interdivision time data on lineage trees to infer the hidden cell cycle factors. (**a**) *Time lapse observations*. Cartoon demonstrating how time-lapse microscopy allows single cells to be tracked temporally as they go through the cell cycle to division. Multiple different factors affect the rate at which cells progress through the cell cycle from birth to subsequent division. *Interdivision time data*. Example lineage tree structure with possible 'family relations' of a cell between which correlations in interdivision time can be calculated. (**b**) *Lineage correlation pattern*. Plot of mother-daughter interdivision time correlation against cousin-cousin interdivision time correlation for the six publicly available datasets used in this work (*Appendix 1— table 1*, *Martins et al., 2018*; *Priestman et al., 2017*; *Chakrabarti et al., 2018*; *Kuchen et al., 2020*; *Mura et al., 2019*). The shaded red area indicates the region where the cousin-mother inequality is satisfied. (**c**) *Identifying hidden cell cycle factors*. Schematic showing the model motivation and process. We produce a generative model that describes the inheritance of multiple hidden 'cell cycle factors' that affect the interdivision time. The model is fitted to lineage tree data of interdivision time, and we analyse the model output to reveal the possible biological factors that affect the interdivision time correlation patterns of cells.

protein expression (*Locke and Elowitz, 2009*, *Figure 1a*, left). This has led to many discoveries in cell cycle dynamics in bacteria (*Taheri-Araghi et al., 2015*; *Martins et al., 2016*; *Martins et al., 2018*; *Kohram et al., 2021*) and mammalian cells (*Barr et al., 2017*; *Arora et al., 2017*; *Ryl et al., 2017*; *Chakrabarti and Michor, 2020*). Early advances included measuring the distribution of division times across single cells (*Powell, 1956*) and the correlations between cellular variables leading to cell size homeostasis (*Taheri-Araghi et al., 2015*), while more recent applications of time-lapse microscopy have captured multiple generations of proliferating cells, making lineage tracing possible (*Errington et al., 2013*; *Cooper and Bakal, 2017*).

While single-cell distributions measure variation between cellular variables, they ignore both temporal signals and variations propagating across generations to entire lineage trees (*Ulicna et al.,*

*2021*; *Kuchen et al., 2020*; *Sandler et al., 2015*; *Cowan and Staudte, 1986*). These lineage tree correlation patterns can be robust and steady, similar to what is known in spatio-temporal pattern formation (*Turing, 1990*; *Chaplain et al., 1999*). Common examples of lineage tree correlation patterns concern the mother-daughter and the sister correlations that have been used to study cell size homeostasis in *E. coli* (*Taheri-Araghi et al., 2015*; *Ho et al., 2018*) and other mechanisms generating correlated interdivision times such as population growth rate (*Powell, 1956*) and initiation of DNA synthesis (*Cooper, 1982*).

A counter-intuitive correlation pattern presented by many cell types is the 'cousin-mother inequality' (*Sandler et al., 2015*), where the interdivision times of cousin cells are more correlated than those of mother-daughter pairs. This inequality can be observed both in bacteria and mammalian cells (*Figure 1b*). More generally, lineage tree data gives rise to correlation patterns by comparing a single cell to any other cell on the tree (*Figure 1a*, right). Family relations – such as daughter, grandmother, cousin cells etc. – encode inheritance patterns, and correlations between these related cells have been used to understand the dynamics of cell populations (*Mohammadi et al., 2021*; *Nakashima et al., 2020*, *Figure 1c*). Several stochastic models have been proposed to explain interdivision time correlation patterns. Most of them make prior assumptions on the underlying mechanism controlling cell division such as those focusing on cell size control (*Ho et al., 2018*), DNA replication (*Cooper and Helmstetter, 1968*; *Cooper, 1982*) or underlying oscillators (*Kohram et al., 2021*). For example, inheritance of DNA content can explain the correlation in interdivision time between sister cells in bacteria (*Cooper, 1982*). Similarly, it has been shown that a simple model with interdivision time correlations (*Cowan and Staudte, 1986*) cannot satisfy the 'cousin-mother inequality' (*Sandler et al., 2015*), but a more complex kicked cell cycle model does (*Mosheiff et al., 2018*). It is presently unclear what information correlation patterns carry about the underlying mechanisms that generate them. This is because a unified and systematic framework to generate any desired interdivision time correlation pattern is lacking.

Here, we propose a stochastic model to investigate how cell cycle factors – which we define in this work as hidden properties that affect interdivision time – shape the lineage tree correlation patterns of cells. These could include physiological factors, such as cell size, growth rate and cell cycle checkpoints, or specific cell cycle drivers such as CDKs, mitogens and division proteins. We will only focus on data describing patterns of interdivision time in bacterial and mammalian cell types, which circumvents intricate measurements of cell volume, mass, and DNA replication. This also avoids dealing with fluorescent reporter strains that may be difficult to engineer depending on cell type. We propose a generative model of correlation patterns that involves a number of hidden cell cycle factors and reduces to common mechanistic cell cycle models for specific parameter choices. Our theory predicts three distinct lineage correlation patterns; aperiodic, alternator and oscillator. We demonstrate how the model can be used to identify these patterns using Bayesian inference in bacteria and mammalian cells. Our analysis reveals several dynamical signatures of cell cycle factors hidden in lineage tree interdivision time data.

## Results

### A general inheritance matrix model provides a unified framework for lineage tree correlation patterns

Previous studies (*Cowan and Staudte, 1986*; *Sandler et al., 2015*) found that simple inheritance rules, where interdivision times are correlated from one generation to another through a single parameter, cannot explain the lineage correlation patterns seen in experimental single-cell data. To address this issue, we propose a unified framework where the interdivision time is determined by a number of *cell cycle factors* that represent hidden variables such as cell cycle phase lengths, protein levels, cell growth rate or other unknowns (*Figure 1c*), that each have their own inheritance pattern.

The states of the cell cycle factors is assumed to be a vector $\boldsymbol{y}_p = (y_{p,1}, y_{p,1}, \ldots, y_{p,N})^\top$ that determine the interdivision time of a cell with index $p$ via

$$\tau_p = f(\boldsymbol{y}_p). \tag{1a}$$

Inheritance from mother to daughter of the $N$ cell cycle factors is described by a nonlinear stochastic Markov model on a lineage tree:

$$
\begin{aligned}
\boldsymbol{y}_{2m} &= \boldsymbol{g}\left(\boldsymbol{y}_m\right) + \boldsymbol{e}_{2m}, \\
\boldsymbol{y}_{2m+1} &= \boldsymbol{g}\left(\boldsymbol{y}_m\right) + \boldsymbol{e}_{2m+1},
\end{aligned}
\tag{1b}
$$

where $m$ in $\mathbb{N}$ denotes the mother cell index and $2m$ and $2m+1$ the daughter cell indices. $f : \mathbb{R}_+^N \to \mathbb{R}_+$ and $\boldsymbol{g} : \mathbb{R}_+^N \to \mathbb{R}_+^N$ are possibly nonlinear functions that model the dependence of the interdivision time on cell cycle factors and the inheritance process. $\boldsymbol{e}_p = (e_{p,1}, e_{p,2}, \ldots, e_{p,N})^\top$ is a noise vector for which the pair $\boldsymbol{e}_{2m}, \boldsymbol{e}_{2m+1}$ are identically distributed random vectors with covariance matrix independent of $m$. A non-zero covariance between these noise vectors can account for correlated noise of sister cells. We implicitly assume symmetric cell division such that the deterministic part of the inheritance dynamics $\boldsymbol{g}$ is identical between the daughter cells. Note that we choose *Equation 1a* to be deterministic since division noise can be modelled by adding one more cell cycle factor that does not affect inheritance dynamics $\boldsymbol{g}$.

The general model (*Equation 1a* and *b*) includes many known cell cycle models as a special case. For example, the interactions between cell cycle factors could model cell size control mechanisms (Appendix 1 - Section A6.1), the coordination of cell cycle phases (Appendix 1 - Section A6.3), or deterministic cues, such as periodic forcing of the cell cycle (Appendix 1 - Section A7.1), or coupling of the circadian clock to cell size control (Appendix 1 - Section A7.2).

The full model can only be solved for specific choices of $f$ and $\boldsymbol{g}$, and these functions are generally unknown in inference problems. To overcome this limitation, we assume small fluctuations resulting in an approximate linear stochastic system (see Appendix 1 - Section A1 for a derivation) involving the interdivision time

$$
\tau_p = \bar{\tau} + \boldsymbol{\alpha}^\top \boldsymbol{x}_p.
\tag{2a}
$$

The vector of cell cycle factor fluctuations $\boldsymbol{x}_p = (x_{p,1}, x_{p,2}, \cdots, x_{p,N})^\top$ obeys

$$
\begin{aligned}
\boldsymbol{x}_{2m} &= \boldsymbol{\theta}\boldsymbol{x}_m + \boldsymbol{z}_{2m}, \\
\boldsymbol{x}_{2m+1} &= \boldsymbol{\theta}\boldsymbol{x}_m + \boldsymbol{z}_{2m+1}.
\end{aligned}
\tag{2b}
$$

Here, $\bar{\tau}$ is the stationary mean interdivision time, $\boldsymbol{\theta}$ is the $N \times N$ inheritance matrix and $\boldsymbol{z}_{2m}$ and $\boldsymbol{z}_{2m+1}$ are two noise vectors of length $N$ that capture the stochasticity of inheritance dynamics and differentiate the sister cells (*Figure 2a*). We denote the $N \times N$ covariance matrices $\boldsymbol{S}_1 = \mathrm{Var}(\boldsymbol{z}_{2m}) = \mathrm{Var}(\boldsymbol{z}_{2m+1})$ and $\boldsymbol{S}_2 = \mathrm{Cov}(\boldsymbol{z}_{2m}, \boldsymbol{z}_{2m+1})$, for all $m$ in $\mathbb{N}$ of the noise terms $\boldsymbol{z}$ (and $\boldsymbol{e}$) in individual cells and between sister cells, respectively. The noise terms are independent for all other family relations. The cell cycle factor fluctuations are scaled such that $\boldsymbol{\alpha} = (\alpha_1, \alpha_2, \ldots, \alpha_N)^\top$ is a binary vector of length $N$ made up of 1 s and 0 s depending on whether the function $f$ determining the interdivision time has dependence on a given cell cycle factor (see Appendix 1 - Section A1 for details). Under this scaling each cell cycle factor has a positive effect on the interdivision time, and hence we do not distinguish between factors with positive or negative effects on interdivision time.

When the special case of a single cell cycle factor ($N = 1$) is considered, the inheritance matrix model system reduces to a well-known model with correlated division times (*Cowan and Staudte, 1986*; *Staudte et al., 1984*; *Staudte, 1992*; *Staudte et al., 1997*), and we will refer to this case as simple inheritance rules (see also Appendix 1 - Section A5). In the following, we will explore the correlation patterns generated by multiple cell cycle factors.

## The inheritance matrix model reveals three distinct interdivision time correlation patterns

Here, we define a correlation pattern to be the correlation coefficients of pairs of cells on a lineage tree. Here we introduce a function $\rho(k, l)$ which we call the *generalised tree correlation function*:

$$
\rho(k, l) = \frac{\mathrm{Cov}(\tau_k, \tau_l)}{s_\tau},
\tag{3}
$$

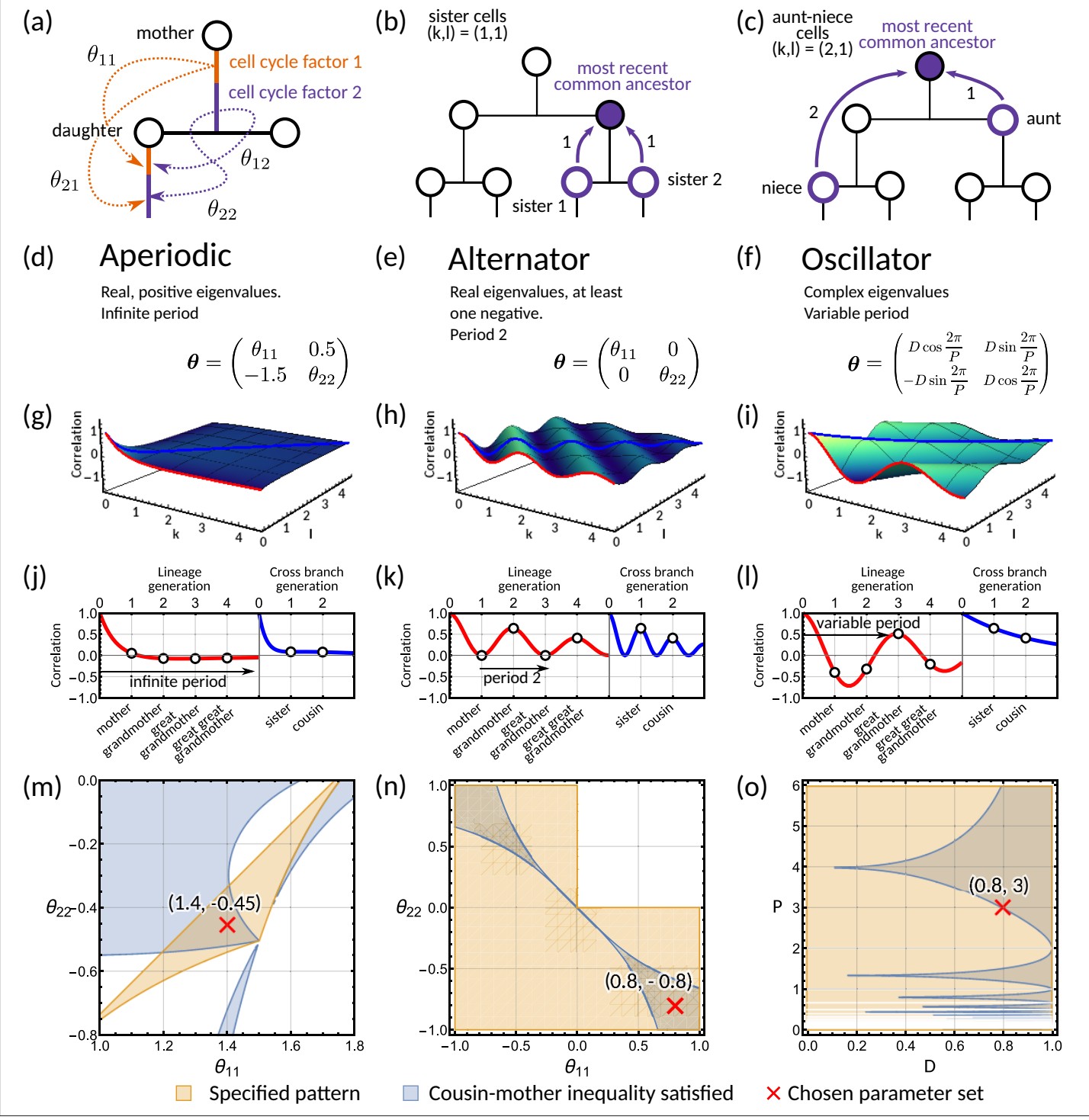

**Figure 2.** Analysis of the inheritance matrix model identifies three distinct lineage tree correlation patterns. (**a**) Diagram illustrating the inheritance matrix model with two cell cycle factors which affect the interdivision time of a cell. Each factor in the mother exerts an influence on a factor in the daughter through the inheritance matrix $\boldsymbol{\theta}$. (**b,c**) Schematics showing how the coordinate $(k, l)$ introduced in 'The inheritance matrix model reveals three distinct interdivision time correlation patterns' is determined. This coordinate describes the distance to the most recent common ancestor for chosen pair of cells. Examples shown are (**b**) sister pairs with $(k, l) = (1, 1)$, and (**c**), aunt-niece pairs with $(k, l) = (2, 1)$. (**d-o**) Panels demonstrating the three correlation patterns that arise from the inheritance matrix model with two cell cycle factors. (**d-f**) Example inheritance matrices $\boldsymbol{\theta}$ that produce the desired patterns: (**d**) aperiodic, (**e**) alternator and (**f**) oscillator correlation patterns. (**g–i**) Three-dimensional plot of the *generalised tree correlation function* (*Equation M3*) demonstrating each of the three patterns. On each plot we highlight the *lineage generation correlation function* ($k = 0$ or

*Figure 2 continued on next page*

*Figure 2 continued*

$l = 0$) (red line) and the *cross-branch generation correlation function* ($k = l$) (blue line). The shading of the 3D plot indicates the correlation coefficient at that point on the surface. (**j–l**) The lineage and cross-branch generation correlation functions plotted individually, showing the different dynamics for each pattern. (**m–o**) Region plots showing parameter values where the relevant pattern is obtained (orange) and where the cousin-mother inequality is satisfied (blue) for the $\theta$ matrices given in panels (**d-f**). White bands on (**o**) indicate where $P = \frac{2}{k}$ which results in real eigenvalues and therefore does not produce an oscillator pattern. Within the parameter region that both produces the desired pattern and also satisfied the cousin-mother inequality, we choose a parameter set (red cross) which is used for the corresponding plots in the panels above. In all panels we fix $\alpha = (1, 1)^T$ and the noise vector $z$ to have covariance equal to the identity matrix.

where $\tau_k$ and $\tau_l$ are the interdivision times of cells in the pair $(k, l)$, and $s_\tau$ is the interdivision time variance. The coordinate $(k, l)$ describes the distance in generations from each cell in the pair to their shared nearest common ancestor (*Figure 2b and c*). We have derived a closed-form formula for $\rho(k, l)$ (*Equation M3*) in Materials and Methods - 'Analytical solution of the inheritance matrix model'; (see Appendix 1 - Section A3 for a full derivation) as a weighted sum of powers of the inheritance matrix eigenvalues $\lambda$:

$$\rho(k, l) = \sum_{i,j=1}^{N} w_{ij} \lambda_i^k \lambda_j^l, \tag{4}$$

with

$$w_{ij} = \frac{\hat{\alpha}_i \hat{\alpha}_j}{\hat{\alpha}^\top \hat{\Sigma} \hat{\alpha}} \left( \frac{(\hat{S}_1)_{ij}}{1 - \lambda_i \lambda_j} + \delta_{k \geq 1} \delta_{l \geq 1} \frac{(\hat{S}_2)_{ij}}{\lambda_i \lambda_j} \right). \tag{5}$$

We observe that the eigenvalues determine the dependence of the tree correlation function on $k$ and $l$, while the noise matrices $S_1$ and $S_2$ determine their relative weights $w_{ij}$ (see *Equation 5*).

Our theoretical analysis reveals three distinct correlation patterns that can be generated by the inheritance matrix model (further details in Materials and methods - 'Analysis of tree correlation patterns'). These can be classified by the eigenvalues of the inheritance matrix $\theta$: (i) if the inheritance matrix exhibits real positive eigenvalues, we observe an *aperiodic* pattern (*Figure 2d*); (ii) if the inheritance matrix has real eigenvalues with at least one negative eigenvalue, we observe an *alternator* pattern (*Figure 2e*); and (iii) if there is a pair of complex eigenvalues we observe an *oscillator* pattern (*Figure 2f*). An intuitive interpretation of the eigenvalue decomposition is that it transforms the cell cycle factors into effective factors inherited independently. Hence, the inheritance matrix is diagonal in this basis. However, the analogy is limited to the case where the inheritance matrix is symmetric and the eigenvalues are real. For simplicity, we will focus on models with two cell cycle factors and note that in higher dimensions ($N \geq 3$), the correlation patterns involve a mixture of the three patterns discussed in detail in this section (*Appendix 1—figure 6c, d and g,h*).

To demonstrate the aperiodic correlation pattern, we utilise an inheritance matrix with positive real eigenvalues (*Figure 2d*). Characteristically, the modelled interdivision time correlations decay to zero as the distance to the most recent ancestor increases (*Figure 2g*) since the eigenvalues in *Equation 4* are bounded between 0 and 1. To look more closely at the patterns on the tree, we utilise two reductions of the generalised tree correlation function. These are the *lineage correlation function* ($\rho(k, l)$ for $k$ or $l = 0$) and the *cross-branch correlation function* ($\rho(k, l)$ for $k = l$). We look at these functions for continuous $k, l$ to visualise better the patterns that occur down the lineage and across the branches of the tree. The lineage correlation function gives the correlation dynamics as you go down the lineage tree, whereas the cross-branch correlation function gives the correlation dynamics as you move across neighbouring branches of the lineage tree. We observe that the interdivision time correlations decrease as we move both across generations and branches (*Figure 2j*).

In contrast, the alternator pattern generates oscillations with a fixed period of two generations in the lineage correlation function. The behaviour is typically observed for cell cycle factors with negative mother-daughter correlations (Appendix 1 - Section A6.1). In this case, we have at least one negative eigenvalue and thus *Equation 4* will alternate between positive and negative values for successive generations, producing the period two oscillation. We demonstrate this correlation pattern for the generalised tree correlation function (*Figure 2h*) using a diagonal $\theta$ matrix (*Figure 2e*). We observe alternating correlations across generations in the lineage correlation function, and the continuous

interpolation of the cross-branch correlation function (*Figure 2k*). Although the period is fixed to two generations, the amplitude of the correlation oscillation varies with the absolute magnitude of the eigenvalues (Materials and methods - 'Analysis of tree correlation patterns').

To investigate the oscillator correlation pattern, we propose a hypothetical inheritance matrix $\theta$ with eigenvalues $\lambda = (De^{+i\frac{2\pi}{P}}, De^{-i\frac{2\pi}{P}})$ which are complex for $D, P \neq 0$ and $P \neq \frac{2}{k}, k$ in $\mathbb{Z}$ (*Figure 2f*). The parameters $P$ and $D$ control the period and the respective damping of an underlying oscillator, i.e., the limit $D \to 1$ leads to an undamped oscillation and $D \to 0$ corresponds to an overdamped oscillation (see Materials and methods - 'Determining the period of correlation oscillations from the eigenvalues' for details). Correspondingly, the graph of the generalised tree correlation function (*Figure 2i*) shows clear oscillations across generations. These correlation oscillations are also evident in the lineage correlation function but are absent in the cross-branch correlation function (*Figure 2l*). However, oscillations are possible in the cross branch correlation function for other choices of $\theta$ with complex eigenvalues (see model fits in Figure 3 and Methods - 'Analysis of tree correlation patterns'). In summary, the qualitative behaviour of the interdivision time correlation patterns can be studied using the eigenvalue decomposition of the inheritance matrix $\theta$.

## The cousin-mother inequality is not required to generate complex correlation patterns

Our analysis shows that of the three specified patterns, only the oscillator pattern cannot arise from simple inheritance rules. This is because it requires at least two inherited cell cycle factors ($N \geq 2$) for the inheritance matrix to possess complex eigenvalues. We therefore asked whether the oscillator pattern is necessary for the cousin-mother inequality to be satisfied. We find that this is not the case, but instead, all three correlation patterns can be compatible with the cousin-mother inequality if $N \geq 2$. To demonstrate this, we choose three specific two-dimensional inheritance matrices $\theta$ that produce the required eigenvalue structure (*Figure 2d–f*). We then use these matrices with our analytical solution for the *generalised tree correlation function* (Materials and methods - 'Analytical solution of the inheritance matrix model') to map the regions where the cousin-mother inequality can be satisfied (*Figure 2m–o*). Interestingly, we find that oscillations can arise even in parameter regions that violate the cousin-mother inequality (*Figure 2o*). We conclude that both the cousin-mother inequality and the oscillator pattern are sufficient but not necessary conditions to rule out simple inheritance rules.

To understand which datasets can be explained by simple inheritance rules, we fit the one-dimensional model ($N = 1$) to six publicly available lineage tree datasets (*Appendix 1—table 1*) using Bayesian methods (Materials and methods - 'Data analysis and Bayesian inference of the inheritance matrix model'). These datasets were chosen as they each had a sufficient number of cells for correlation analysis and covered a broad range of cell types. We found that the model fit is poor for the datasets that display the cousin-mother inequality, which is the case for cyanobacteria, clock-deleted cyanobacteria, neuroblastoma and human colorectal cancer cells (*Appendix 1—figure 1a–f*). Despite not obeying the cousin-mother inequality, the fit is also poor for mouse embryonic fibroblasts (*Appendix 1—figure 1f*) as the median inferred correlation lies outside the 95% confidence intervals for both the grandmother and cousin correlations which are included in the model fit, and the confidence intervals for the data vs the credible intervals from the inference show minimal overlap (*Appendix 1—figure 2f*). Another inequality may be violated in this dataset that cannot be explained using the one-dimensional model, suggesting that the absence of the cousin-mother inequality cannot rule out more complex division rules. The only cell type that has a good fit for the one-dimensional model is mycobacteria (*Appendix 1—figure 1c*). We thus conclude that the majority of the datasets must be described by higher dimensional inheritance dynamics of multiple cell cycle factors.

## The two-dimensional inheritance matrix model fits interdivision time correlation patterns from a range of cell types

We asked whether the correlation patterns are better described by a two-dimensional inheritance matrix model. Bayesian inference (Materials and methods - 'Data analysis and Bayesian inference of the inheritance matrix model') produced a good model fit for all six datasets (*Figure 3a–f*) for the two-factor inheritance matrix model, within relatively narrow error bars of mother, grandmother, sister and cousin correlations (*Appendix 1—table 1*). The credible intervals from the Bayesian inference matched the confidence intervals of correlations used for fitting (*Appendix 1—figure 2*). We

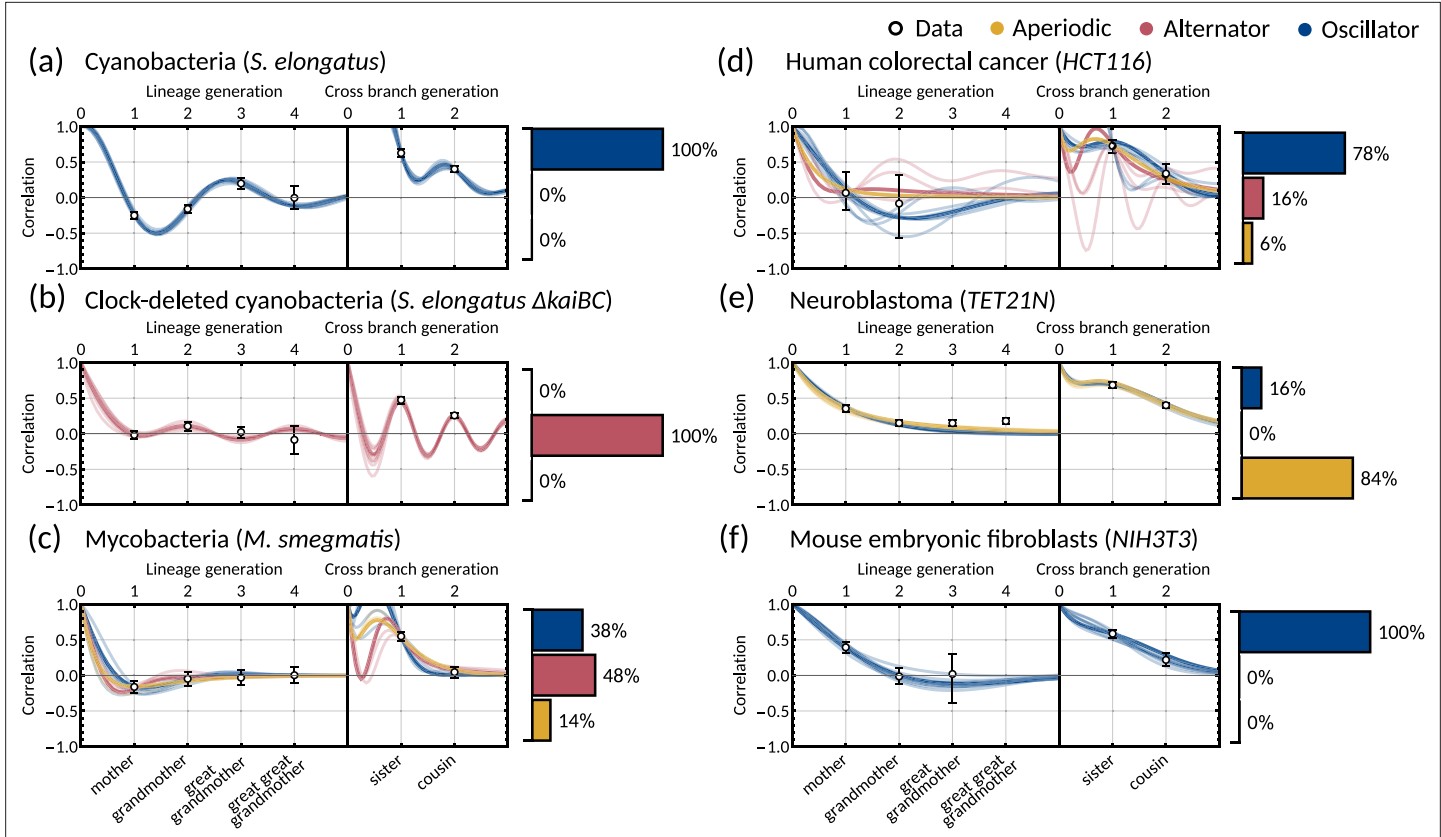

**Figure 3.** The inheritance matrix model with two cell cycle factors fits interdivision time correlation patterns for a range of cell types. Posterior correlation functions based on fitting to mother-daughter, grandmother-granddaughter, sister-sister and cousin-cousin correlations for three bacterial (left) and three mammalian (right) datasets: (**a**) cyanobacteria, (**b**) clock-deleted cyanobacteria, (**c**) mycobacteria, (**d**) human colorectal cancer, (**e**) neuroblastoma, and (**f**) mouse embryonic fibroblasts. Pearson correlation coefficients (white circles) and 95% bootstrapped confidence intervals (error bars) obtained through re-sampling with replacement of the original data (10,000 re-samples). Posterior distribution samples were clustered into aperiodic, alternator, and oscillator patterns (bar charts). We show multiple representative samples (solid and shaded lines) drawn from the posterior distribution *Appendix 1—figure 2* without clustering. Where correlations appear missing, this is in cases where the lineage trees in the data were not deep enough for the correlations to be calculated. Only lineage and cross branch generations 1 and 2 were used in model fitting. Here all panels assume $\boldsymbol{\alpha} = (1, 1)^{\top}$, but taking $\boldsymbol{\alpha} = (1, 0)^{\top}$ produces similar results (*Appendix 1—figure 4*).

quantified the quality of our fits using the Akaike information criterion (AIC) (Materials and methods - 'Data analysis and Bayesian inference of the inheritance matrix model', (*Equation M11*)) for each dataset and compared these to the one-dimensional model (*Appendix 1—table 1*). The AIC estimates the goodness of fit with a penalty for model complexity allowing us to select the simplest model that explains the data. The AIC values indicate that the inheritance matrix model with two cell cycle factors provides the simplest fit for all cell types used here, except for the mycobacteria data where simple inheritance rules provided an equally good fit with a significant reduction in the number of model parameters. We expected the AIC to select the two dimensional model where the cousin-mother inequality was satisfied such as in cyanobacteria, clock-deleted cyanobacteria, neuroblastoma, and human colorectal cancer cells. The match with the two-factor inheritance matrix model in fibroblasts was less obvious.

Crucially, we find that the model has a good predictive capacity for correlations further down the lineage tree. For each pattern, we show several samples from the conditional posterior distribution (solid and shaded lines) to illustrate fits of the lineage correlation and cross-branch correlation function (*Figure 3a–f*). For all datasets except neuroblastoma, the curves also intercept the great-grandmother and great-great-grandmother correlations that were not used for fitting (*Figure 3a–d and f*), and bootstrapped confidence intervals from the data overlapped with the credible intervals obtained from Bayesian inference (*Appendix 1—figure 2*). We then asked which correlation patterns underlie the data. To assess this, we calculated the eigenvalues of each posterior sample of the inheritance

matrix to categorise the aperiodic, alternator and oscillator patterns (*Figure 3a–f*, bar charts). We found that in every dataset, the dominant correlation pattern was identifiable with probabilities well above 50%, except for mycobacteria (*Figure 3c*) that was better described by simple inheritance rules (*Appendix 1—figure 1c*).

Cyanobacteria, (*Figure 3a*), human colorectal cancer (*Figure 3d*), and mouse embryonic fibroblasts (*Figure 3f*) display a dominant oscillator pattern, but we see that their lineage correlation functions exhibit widely different periodicities. For example, the posterior lineage correlation for cyanobacteria displays a higher frequency oscillation than those in human colorectal cancer cells and fibroblasts. Clock-deleted cyanobacteria (*Figure 3b*) and mycobacteria (*Figure 3c*) display a dominant alternator pattern which could be induced by strong sister correlations. We see that clock-deleted cyanobacteria (*Figure 3b*) has a 100% alternator pattern in contrast to the 100% oscillator pattern seen for wild type cyanobacteria, suggesting that the deletion of the clock gene has completely transformed the correlation pattern and has abolished the underlying oscillation. Neuroblastoma (*Figure 3e*) displays a dominant aperiodic pattern. The predictive capacity for this cell type is weaker than for the other datasets, which we assume is due to the tight confidence interval in the correlations. Despite this discrepancy, we find that the inheritance matrix model produces excellent fits and has good predictive capacity for all other cell types studied in this work.

## Bayesian inference reveals that individual inheritance parameters are not identifiable

We next ask which mechanisms are responsible for generating the observed correlation patterns. The Bayesian inference used for model fitting (Materials and methods - 'Data analysis and Bayesian inference of the inheritance matrix model') samples parameters using a MCMC Gibbs sampler. The Gibbs sampler can be thought of as a random walk in parameter space that settles around parameter regions with high likelihood. We found that the explorations of the Gibbs sampler did not settle in a particular parameter subspace but meandered off to explore vast areas of the parameter space without improving the likelihood values (*Appendix 1—figure 3a and b*). Such behaviour is expected when model parameters are not identifiable and the posterior distribution of parameters cannot be efficiently sampled (*Rannala, 2002*; *Raue et al., 2013*).

To provide further evidence of unidentifiablity, we obtained four histograms of a single parameter of the inheritance matrix for different initialisations. The four distributions are very different (*Figure 4a*), showing that the random walk does not settle to a stationary distribution. We further observe that the mean squared displacement increases without bound (*Figure 4b*) showing that the sampling does not settle in a particular subset of the parameter space. In contrast to the individual parameters, the sampled posterior distribution of the eigenvalues is consistent across the averages (*Figure 4c*) and their mean squared displacement converges rapidly (*Figure 4d*). We note that unidentifiability arises for the inheritance matrix model with multiple cell cycle factors and does not feature for simple inheritance rules (Appendix 1 - Section A5). This ultimately demonstrates that the interdivision time correlation patterns do not identify a single set of inheritance parameters, but rather need to be described by a distribution of inheritance mechanisms.

## The inheritance matrix model predicts the hidden dynamical correlations of cell cycle factors

Clock-deleted cyanobacteria and neuroblastoma both satisfy the cousin-mother inequality (*Figure 1b*), which indicates that at least two cell cycle factors are responsible for the corresponding correlation patterns. The eigenvalues of the inheritance matrix concentrate in different regions of the admissible parameter space (*Figure 4e*), suggesting the correlation patterns that generate the cousin-mother inequality are distinct. For the clock-deleted cyanobacteria dataset, we found that all posterior samples were consistent with an alternator correlation pattern, while most posterior samples presented aperiodic correlation patterns in neuroblastoma (*Figure 3b and e* bar charts).

We hypothesised that different inheritance models generate these patterns. To verify this hypothesis and since we cannot identify the cell cycle factors directly, we computed the mother-daughter correlations between the two hidden cell cycle factors. Since the order of factors is interchangeable, we only distinguish between mother-daughter correlations between the same ($\mathrm{corr}(x_{m,i}, x_{2m,i})$ and $\mathrm{corr}(x_{m,i}, x_{2m+1,i})$ for $i = 1, 2$) and alternate factors ($\mathrm{corr}(x_{m,i}, x_{2m+1,j})$ and $\mathrm{corr}(x_{m,i}, x_{2m,j})$ for $i \neq j = 1, 2$).

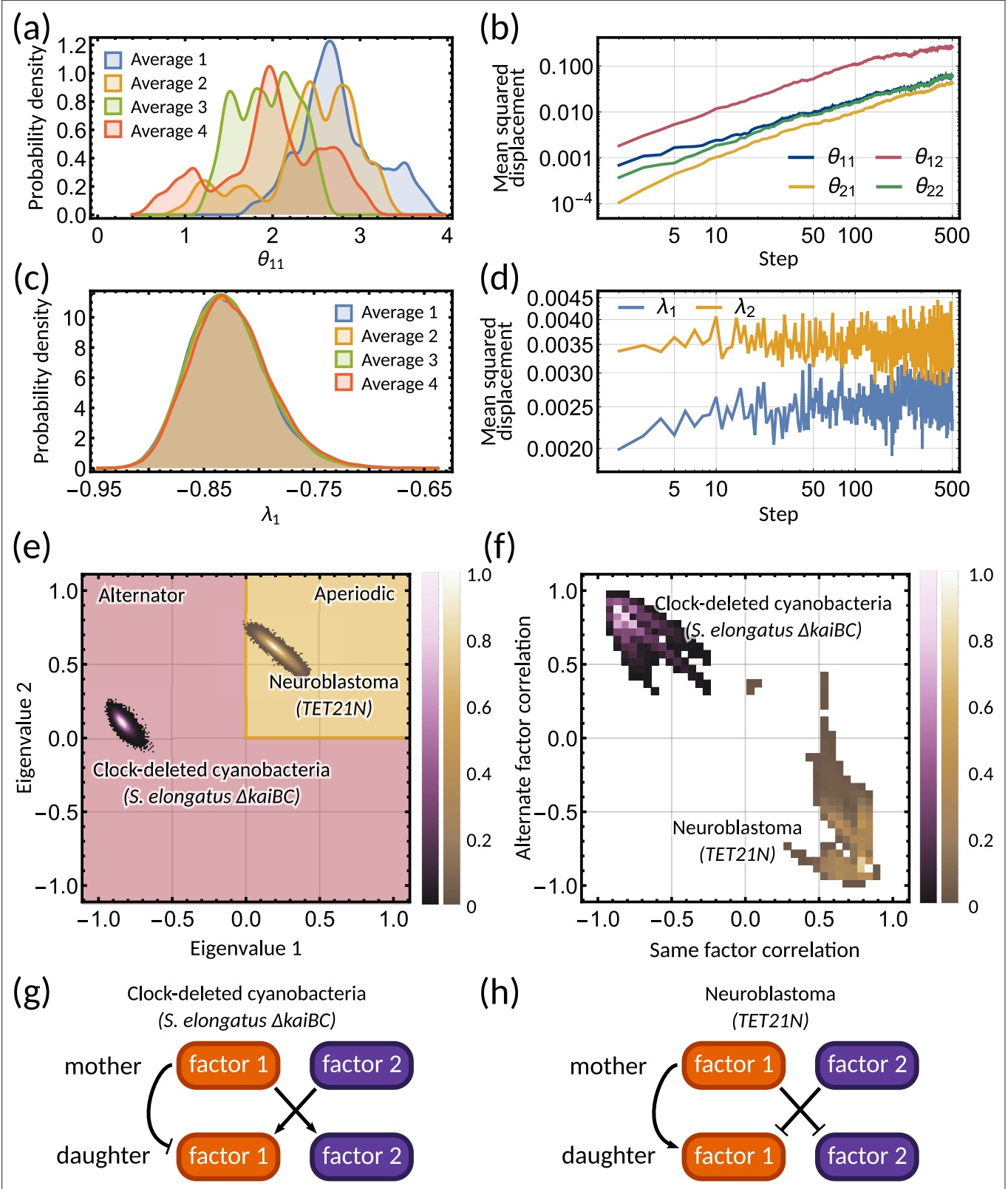

**Figure 4.** Bayesian inference predicts hidden dynamical correlations between cell cycle factors. (**a**) Posterior distribution histograms for $\theta_{11}$ depend on the realisations of a Gibbs sampler and do not settle to a stationary distribution. (**b**) A log-log plot of mean squared displacement for the four $\theta$ variables that make up the inheritance matrix $\boldsymbol{\theta}$. The mean squared displacement for all four parameters increases linearly, meaning the sampling does not settle in any particular region of parameter space. (**c**) Sampled posterior distribution histograms for the eigenvalue $\lambda_1$ for each realisation. The

*Figure 4 continued on next page*

*Figure 4 continued*

histograms are almost identical across the four averages, showing the distribution has converged. (**d**) Mean squared displacement for the eigenvalues of the inheritance matrix $\boldsymbol{\theta}$ settles to a finite value. Plots (**a**) - (**d**) utilise sampling from the inference for the clock-deleted cyanobacteria dataset. (**e**) Density histogram of the real eigenvalue pairs for clock-deleted cyanobacteria (pink) and neuroblastoma (brown) demonstrating where the eigenvalues lie in the aperiodic (yellow) and alternator (red) regions. (**f**) Density histogram of same-factor against alternate-factor mother-daughter correlation for clock-deleted cyanobacteria (pink) and neuroblastoma (brown). We take a minimum threshold of 0.3 for the probability density to remove irrelevant samples. (**g–h**) Influence diagrams for same factor vs alternate factor correlations for (**g**) clock-deleted cyanobacteria and (**h**) neuroblastoma.

The resulting posterior distributions revealed distinct correlation patterns of cell cycle factor correlations for clock-deleted cyanobacteria and neuroblastoma (*Figure 4f*). For clock-deleted cyanobacteria, we predict that at least one factor has a negative mother-daughter correlation while its cross-correlation with the other factor must be positive; while the correlations are of opposite sign for neuroblastoma (*Figure 4f*). We sketch influence diagrams that summarise these relationships between factors (*Figure 4g and h*). Thus, the different interdivision time correlation patterns observed for clock-deleted cyanobacteria and neuroblastoma stem from distinct hidden correlation patterns of cell cycle factor fluctuations.

## The inheritance matrix model reveals biological rhythms underlying the cell cycle

We observe that the lineage correlation functions of cyanobacteria, human colorectal cancer cells, and fibroblasts exhibit vastly different correlation oscillation periods (*Figure 3*). Next, we are interested to see whether the oscillations seen in these datasets are compatible with biological oscillators known to affect cell cycle control.

### Correlation oscillations and underlying rhythms can exhibit vastly different periods

The period of the correlation oscillation is related to the location of the eigenvalues of the inheritance matrix on the complex plane. We consider an eigenvalue $\lambda$ of the inheritance matrix. In terms of the mean interdivision time $\bar{\tau}$, the correlation period $T_0$ is:

$$T_0 = \bar{\tau} \frac{2\pi}{|\mathrm{Arg}(\lambda)|} \geq 2\bar{\tau}, \tag{6}$$

and the inequality means that the period $T_0$ is always greater than twice the mean interdivision time $\bar{\tau}$. More generally, there is an oscillation period associated with each eigenmode of the inheritance matrix, but the period is infinite for real eigenvalues, and thus only complex eigenvalues generate correlation oscillations. This inequality follows from *Equation 6* using $|\mathrm{Arg}(\lambda)| \leq \pi$. However, known biological oscillators that influence cell cycle control often have periods *less* than twice the mean interdivision time, such as stress response regulators (*Harper et al., 2018*; *Stewart-Ornstein et al., 2017*) and gene expression oscillations (*William et al., 2007*; *Gao et al., 2014*; *Whitfield et al., 2002*). How can relatively slow observed correlation oscillations be compatible with much faster biological oscillators underlying the cell cycle?

The resolution to this issue is that the period of the correlation oscillation does not always match the frequency of the underlying oscillator. Instead there are a number of possible oscillator periods $T_n$ compatible with the correlation oscillation period $T_0$ (Appendix 1 - Section A4) given by:

$$T_n = \frac{\bar{\tau} T_0}{|\bar{\tau} + n T_0|}, \tag{7}$$

for $n$ in $\mathbb{Z}$. This phenomenon, that the same correlation oscillation can be explained by multiple underlying oscillators, can be understood using the intuition in *Figure 5a*.

### Circadian oscillations in cyanobacteria and fibroblasts support coupling of the circadian clock and the cell cycle

Cyanobacteria and fibroblasts both exhibit correlation patterns consistent with an oscillator underlying cell divisions (*Figure 3e*, bar chart). We observe that the posterior distribution of the eigenvalues

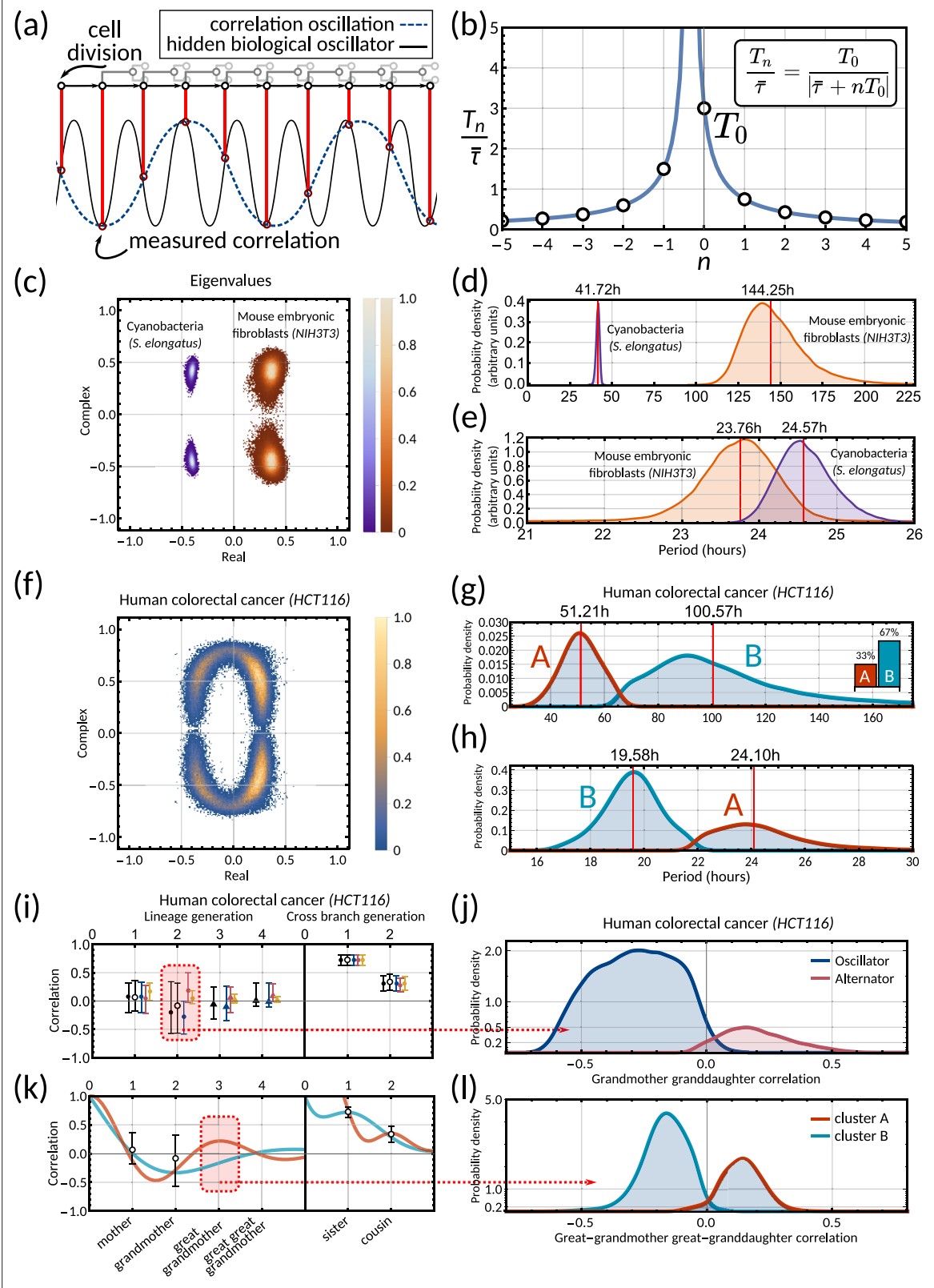

**Figure 5.** The inheritance matrix reveals the periodicity of hidden biological oscillators underlying the cell cycle. (**a**) Schematic showing how sampling a high frequency rhythm at each cell division could result in a lower frequency oscillator being constructed. (**b**) Possible oscillator periods (*Equation 7*) indexed by $n$ for a correlation oscillation period $T_0 = 3\bar{\tau}$. (**c**) Density plot of the complex eigenvalue output from the model sampling for cyanobacteria (purple) and mouse embryonic fibroblasts (orange). (**d**) Posterior distributions of the correlation oscillation period $T_0$ in cyanobacteria (purple) and

*Figure 5 continued on next page*

*Figure 5 continued*

mouse embryonic fibroblasts (orange). (**e**) Posterior distributions of the oscillator period $T_{-1}$ in cyanobacteria (purple) and mouse embryonic fibroblasts (orange). Arbitrary units in (**d**) and (**e**) are used to compare histograms, the density values are not normalised in relation to each other in order to display both histograms clearly on the same plot. (**f**) Density plot of complex eigenvalues for human colorectal cancer. (**g**) Posterior distributions of the correlation oscillation period in human colorectal cancer (shaded area) and oscillator clusters corresponding to positive (cluster A, orange) and negative real parts (cluster B, blue). The bar chart shows the posterior mass of the clusters. (**h**) Posterior distributions of the oscillator periods $T_{-1}$ corresponding to (**g**). (**i**) Model fit and 95% credible intervals for human colorectal cancer (cf. legend of *Figure 3*). Red area indicates the grandmother granddaughter correlation explored in (**j**). (**j**) Posterior distribution of oscillator vs alternator clusters give grandmother correlations with opposite signs. (**k**) Lineage and cross-branch correlation functions of oscillator clusters A (orange) and B (blue) in human colorectal cancer. Red area indicates the great-grandmother great-granddaughter correlation explored in (**l**). (**l**) Posterior distributions of oscillator clusters A (orange) and B (blue) have great-grandmother correlations of opposite signs.

is confined to a region with negative real parts for cyanobacteria and positive real parts for fibroblasts (*Figure 5c*). Using these distributions, we estimate the median period of the correlation oscillations (using *Equation 6*) to be 41.7 hr for cyanobacteria and 144.3 hr for fibroblasts (*Figure 5d*). We wondered whether the stark difference in the periods of the correlation oscillations indicates a different underlying rhythm. Conversely, we found this was not the case, but both correlation patterns were consistent with an approximate circadian rhythm. The posterior of the oscillator period $T_{-1}$, which is closest to the period of correlation oscillation $T_0$, suggests a median period of 24.6 hr for cyanobacteria and a median period of 23.8 hr for fibroblasts (*Figure 5e*). We also validated the inference result using simulated data (*Appendix 1—figure 9*). This finding supports a strong coupling of circadian rhythms to the cell cycle, as reported previously for both cyanobacteria (*Yang et al., 2010*; *Martins et al., 2018*) and fibroblasts (*Nagoshi et al., 2004*; *Menger et al., 2007*; *Nagoshi et al., 2005*). Notably, we see that clock-deleted cyanobacteria displays 100% alternator pattern (*Figure 3b*) and therefore has a lineage tree correlation pattern that cannot be described by an approximate 24 hr oscillator, in contrast to wild type cyanobacteria.

## Bimodal posterior distribution of underlying oscillations in human colon cancer

Finally, we turn to the analysis of cancer cell data. The dominant correlation pattern was oscillatory (78% posterior probability, *Figure 3d*, bar chart). The posterior distribution of complex eigenvalues for the oscillator pattern has support in a large region of the parameter space. It has two distribution modes depending on whether the eigenvalues have positive or negative real parts (*Figure 5f*). Similarly, the posterior of the correlation oscillation period is bimodal, too (*Figure 5g*), which means that two competing oscillator patterns are compatible with the data.

To disentangle these alternative hypotheses, we cluster the posterior samples by the real part of the eigenvalues. We label cluster A for negative real parts and cluster B for positive ones. The correlation periods of the individual clusters do not provide us with immediate clues about the underlying oscillators. Cluster A has a median correlation oscillation period of 51.2 hr while cluster B has a median period of 100.6 hr (*Figure 5g*). We therefore inspected the oscillator periods $T_{-1}$ for each cluster, which are closest to the observed correlation period (*Figure 5h*). The median of the predicted oscillator period of cluster A has an oscillator period $T_{-1}$ of 24.1 hr, which hints at a circadian oscillator underlying the cell cycle in agreement with a previous model (*Chakrabarti et al., 2018*). However, only about 33% of posterior samples with complex eigenvalues were assigned to this cluster. The majority of posterior samples, cluster B, had a different predicted period with a median of 19.6 hr (*Figure 5h*). A possible explanation is that the circadian period is shortened in cancer cells.

A strength of the Bayesian framework is that it allows us to express our confidence in this prediction. We find that our analysis is not conclusive about the correlation pattern as 78% of posterior samples showed an oscillator pattern. As a result, about 52% of all the posterior samples favour a 19.6 hr oscillator and 26% for the 24.1 hr oscillator, matching approximately circadian rhythm. 16% of the samples demonstrate alternator correlation patterns, and the remaining 6% samples are aperiodic (compare bar charts in *Figures 3d and 5g*). We therefore ask whether these competing models make predictions that translate into testable hypotheses. We found that the oscillator correlation pattern predicts a negative grandmother correlation while the alternator pattern predicts a positive grandmother correlation (*Figure 5i and j*). Thus, measuring the grandmother correlation with

higher precision, for example, via increasing sample size, would tighten the confidence intervals of measured correlations (*Figure 5i*), and improve our ability to narrow down the true pattern. On the contrary, predicting the great-grandmother correlation allows us to distinguish between the 19.6 hr and 24.1 hr rhythms (*Figure 5h*). Posterior samples in cluster A predicted a positive interdivision time correlation between a cell and its great-grandmother, while cluster B predicted a negative correlation (*Figure 5k and l*). While the great-grandmother correlation could not be estimated using the present data, deeper lineage trees could be used to discriminate the period of the biological oscillator and help reveal whether the circadian period is altered in cancer cells, or not. In summary, our theory helps to predict the hidden periodicities of biological oscillators from lineage tree interdivision time data.

## Discussion

We propose a Bayesian approach to predict hidden cell cycle factor dynamics from interdivision time correlation patterns. Our underlying model fits the lineage tree data for a range of bacterial and mammalian cell types and allows us to classify different correlation patterns. Our inference demonstrates that these patterns are identifiable, but the individual inheritance parameters are not. This finding suggests that interdivision time correlations alone are insufficient to gain mechanistic insights into cell cycle control mechanisms. The identified correlation patterns, however, reveal the dynamics of the underlying cell cycle factors.

We focused on a data-driven approach without any prior assumptions of the division mechanism, allowing the interdivision time data to speak for itself. Other studies used a model similar to the inheritance matrix model proposed here, and linked latent factors to the interplay between cell cycle progression and growth (*Kuchen et al., 2020*). Auto-regressive models have also been used in bacteria to discriminate between different mechanisms of cell size control (*Kohram et al., 2021*). Additionally, they have been used to combine growth and cell cycle reporters to explain interdivision time dynamics in fibroblasts (*Mura et al., 2019*). In principle, the inheritance matrix model can be used to model the inheritance dynamics of any factor affecting the interdivision time of a cell. In fact, it comprises many mechanistic models as special cases, such as those based on DNA replication, cell size control or cell cycle phases (Appendix 1 - Section A6 and *Appendix 1—figures 5 and 6*). In future work, it will be useful to improve the identifiability of the model parameters. This could be accomplished either through including knowledge of inheritance mechanisms through prior distributions, or by including additional data on measured cell cycle factor dynamics – such as cell cycle phases, cell size, protein expression etc. – in the inference.

Another limitation of our inference is that we computed the interdivision time variance $s_\tau$ in *Equation M2* of the model assuming that trees have equal number of generations in each branch. The advantage of this estimator is that it does not assume any particular noise distribution but this may lead to a statistical bias compared to the sample variance of tree-structured data with branches of varying length (*Powell, 1956*; *Hashimoto et al., 2016*; *Thomas, 2017*; *Jafarpour et al., 2018*; *Kuchen et al., 2020*; *Sandler et al., 2015*). However, the approximation does not change the identified correlation patterns and the conclusion of this work, since any variance bias can be compensated by multiplying the noise matrices ($S_{1,2}$ in *Equation 5*) with a constant, and, for the data analysed, the interdivision time variance estimators cannot be distinguished within the 95% confidence intervals (*Appendix 1—table 3*). Developing a theory correcting for such biases in lineage tree data will be the subject of future work.

An important result of the present analysis is that lineage tree correlation patterns of very different cell types – cyanobacteria, mouse embryonic fibroblasts and human colorectal cancer – can be explained through an underlying circadian oscillator coupled to cell division. While the coupling between the cell cycle and circadian clock is well established both in cyanobacteria and mouse embryonic fibroblasts, it is less well studied in cancer (*Shostak, 2017*; *Kiessling et al., 2017*). Our method robustly reconstructs the circadian rhythms from the interdivision time correlation patterns despite the lack of the cousin-mother inequality for fibroblasts, demonstrating the cousin-mother inequality is not required for complex correlation patterns ('The cousin-mother inequality is not required togenerate complex correlation patterns'). It is interesting to observe the differences in the oscillatory correlation patterns in these organisms. They are characterised by complex eigenvalues with negative real parts in cyanobacteria, but positive real parts in fibroblasts (*Figure 5c*), resulting in opposite mother-daughter correlations for these datasets (*Figure 3a and f*).

It would be interesting to explore what mechanisms underlie these different patterns. While the circadian clock in fibroblasts relies on transcriptional mechanisms (*Hughes et al., 2009*; *Menger et al., 2007*; *Takahashi, 2017*), the origin of the clock is non-transcriptional in cyanobacteria (*Cohen and Golden, 2015*; *Tomita et al., 2005*; *Nakajima et al., 2005*). The negative mother-daughter correlation in cyanobacteria likely stems from size control mechanisms that are modulated by the circadian clock (*Martins et al., 2018*). However, the mechanisms that generate positive mother-daughter correlations in fibroblasts are still to be explored. Interestingly, in human colorectal cancer, two oscillatory correlation patterns divide the posterior distributions into two distinct clusters with positive and negative mother-daughter correlations. If the circadian clock was to generate a positive mother-daughter correlation, as it does in fibroblasts which have a structurally related clock, the period corresponds to a 20 hr rhythm. This finding thus suggests that the circadian period is altered in cancerous cells. Indeed, several studies report similar periods of 18 hr and 20 hr for gene expressions in the human colorectal cancer core-clock (*Fuhr et al., 2019*; *Parascandolo et al., 2020*).

Our theory predicts that an oscillator's period does not always match the period of the observed correlation oscillations. We describe a lower bound on the correlation period that is reminiscent of the Nyquist-Shannon sampling theorem. This theorem describes temporal aliasing in digital audio processing, where a high-frequency signal produces low-frequency oscillations when sampled at a frequency less than twice the sampling frequency. Similarly, spatial aliasing is observed in digital image processing as a moire pattern. In our analogy, the high-frequency signal is a biological oscillator that couples to cell division and is sampled at the cell division frequency (*Figure 5a*). Our result thus extends the Nyquist-Shannon sampling theorem to lineage trees. Our finding has fundamental implications for the reconstruction of oscillator periods from interdivision time data, revealing that there exists a number of oscillators that can all explain the same correlation pattern.

Here, we concentrated on the oscillator periods $T_{-1}$ that are closest to the correlation oscillation periods $T_0$. In principle, we cannot exclude that oscillators with shorter physiological periods are contributing to the observed lineage tree correlation patterns. For example, HES1 expression oscillates with a period of around 5 h in human colon cancer cells (*William et al., 2007*; *Gao et al., 2014*). The stress response regulators NF-κB and p53, which are critical for tumour development, oscillate with periods of approximately 100 min and 5 hr, respectively (*Harper et al., 2018*; *Stewart-Ornstein et al., 2017*). The posterior distributions for periods in this region are not well separated (*Appendix 1—figure 7c*), which makes it challenging to identify factors that oscillate significantly faster than the cell cycle using interdivision time data. It is, however, unknown whether such hypothetical factors couple to cell division specifically in a manner to induce oscillatory interdivision time correlation patterns.

Going forward, there is a need to go beyond the Nyquist-Shannon limit and develop methods that have increased sensitivity to discriminate a broader range of oscillator periods. One way to circumvent the limitation would be to employ fluorescent reporters of the circadian clock that could be correlated directly with cell division timing. Another way, would be to provide parallel readouts of the underlying rhythm through events that sub-sample the cell cycle, such as DNA replication, or the timing of individual cell cycle phases. Not only would we be able to look at the correlation in interdivision time between cells on a lineage tree, but we would also be able to analyse the correlations between individual phases and family members, to reveal specific phase control mechanisms. Our main findings result from the the inheritance matrix model with two cell cycle factors, as this was sufficient to explain the correlation patterns of the chosen data. In principle, increasing the number of interacting cell cycle factors can lead to more complex composite patterns that involve combinations of the three patterns discussed in this paper, such as the alternator-oscillator (*Appendix 1—figure 6c and d*), aperiodic-oscillator (*Appendix 1—figure 6g and h*), or birhythmic correlation patterns. Such composite patterns could also arise as the result of nonlinear fluctuations that, within our framework, can be described by adding complexes of cell cycle factors to the inheritance matrix model (Appendix 1 - Section A2). The presence of such complexes induces higher-order harmonics in the correlation oscillations, similar to those observed in the cyanobacterial and mammalian circadian clock (*Martins et al., 2016*; *Thomas et al., 2013*), and detecting such complexes could provide an alternative route to increase the sensitivity of our inference method.

In summary, our findings highlight the predictive power of Bayesian inference on single-cell data and how it can be leveraged to draw testable hypotheses for the design of future experiments. This

was exemplified for human colorectal cancer cells, where various patterns were compatible with the data, something that non-probabilistic approaches cannot accomplish as they fit only a single correlation pattern. In the future, it will be crucial to understand why different cell types have evolved specific lineage correlation patterns and how these patterns affect cell proliferation and disease. It would be interesting to understand whether specific correlation patterns give or reveal some fitness advantage and whether we can use them to predict cell survival. We anticipate that identifying hidden cell cycle factors and their rhythmicity using non-invasive methods such as interdivision time measurements will be instrumental in answering these questions and may benefit other fields where cell proliferation plays a pivotal role.

## Code availability

Code available at https://github.com/pthomaslab/Lineage-tree-correlation-pattern-inference (copy archived at swh:1:rev:dc69bbce5ce909813d7d4356c9fd2da045e02c79; *Hughes, 2022*).

## Materials and methods
### Analytical solution of the inheritance matrix model

From *Equation 2b* and $\mathrm{E}[z_p] = 0$, for all $p$ in $\mathbb{N}$, we see that the vector of cell cycle factors has zero mean $\mathrm{E}[x_P] = \mathbf{0}$. Its $N \times N$ covariance matrix $\mathbf{\Sigma} = \mathrm{Cov}(x_p, x_p)$ satisfies a discrete-time Lyapunov equation:

$$S_1 = \mathbf{\Sigma} - \boldsymbol{\theta}\mathbf{\Sigma}\boldsymbol{\theta}^{\top}. \tag{M1}$$

From the solution of *Equation M1*, we compute the variance of the interdivision time

$$s_\tau = \boldsymbol{\alpha}^{\top}\mathbf{\Sigma}\boldsymbol{\alpha}, \tag{M2}$$

and the generalised tree correlation function $\rho(k, l)$ (see Appendix 1 - Section A3 for a detailed derivation) given by:

$$\rho(k, l) = \frac{\boldsymbol{\alpha}^{\top}\boldsymbol{\omega}(k, l)\boldsymbol{\alpha}}{\boldsymbol{\alpha}^{\top}\mathbf{\Sigma}\boldsymbol{\alpha}}, \tag{M3}$$

where $\boldsymbol{\omega}(k, l) = \boldsymbol{\theta}^k \mathbf{\Sigma} \left(\boldsymbol{\theta}^l\right)^{\top} + \delta_{k \geq 1}\delta_{l \geq 1}\boldsymbol{\theta}^{k-1}S_2\left(\boldsymbol{\theta}^{l-1}\right)^{\top}$ with

$$\delta_{i \geq 1} = \begin{cases} 1 & \text{if } i \geq 1 \\ 0 & \text{otherwise} \end{cases} \quad \text{for } i = k, l. \tag{M4}$$

To ensure that the lineage tree correlation pattern is stationary, we require $\mathrm{SR}(\boldsymbol{\theta}) < 1$ where $\mathrm{SR}(\boldsymbol{\theta}) = \max(\lambda_1, \lambda_2, \ldots, \lambda_N)$ is the spectral radius of $\boldsymbol{\theta}$. This also ensures that the solutions to *Equation M1*; $\mathbf{\Sigma}$, $S_1$ and the function *Equation M3* are unique and independent of the initial conditions.

### Analysis of tree correlation patterns

The patterns of the generalised tree correlation function can be characterised through its eigendecomposition. The general decomposition proceeds through finding the matrix of eigenvectors $U$ of $\boldsymbol{\theta}$ such that

$$U\boldsymbol{\theta}U^{-1} = \mathrm{diag}(\lambda_1, \lambda_2, \ldots, \lambda_N) \tag{M5}$$

is the diagonal matrix of eigenvalues. Defining $\hat{S}_{1,2} = US_{1,2}U^{\top}$ and $\hat{\boldsymbol{\alpha}} = (U^{-1})^{\top}\boldsymbol{\alpha}$, the solution to *Equation M1* is given by

$$\mathbf{\Sigma}_{ij} = \sum_{k,l=1}^{N} U_{ik}^{-1} U_{jl}^{-1} \frac{(\hat{S}_1)_{kl}}{(1 - \lambda_k \lambda_l)}. \tag{M6}$$

This result can then be used to find an explicit expression for the generalised tree correlation function:

$$\rho(k,l) = \sum_{i,j=1}^{N} \frac{\hat{\alpha}_i \hat{\alpha}_j}{\hat{\alpha}^\top \hat{\mathbf{\Sigma}} \hat{\alpha}} \hat{\omega}_{ij}(k,l), \tag{M7}$$

where

$$\hat{\omega}_{ij}(k,l) = \frac{(\hat{\mathbf{S}}_1)_{ij} \lambda_i^k \lambda_j^l}{(1 - \lambda_i \lambda_j)} + \delta_{k\geq 1} \delta_{l\geq 1} (\hat{\mathbf{S}}_2)_{ij} \lambda_i^{k-1} \lambda_j^{l-1}. \tag{M8}$$

*Equation M7* can be rewritten as a superposition of patterns *Equation 4* with weights given by *Equation 5*.

The pattern of the tree correlation function is thus governed by the eigenvalues of the inheritance matrix $\theta$: (i) if one eigenvalue, say $\lambda_1$, is positive then the factor $\hat{\omega}_{11}(k,0) = \hat{\omega}_{11}(0,k) \propto \lambda_1^k$ contributing to lineage correlation decays monotonically. The factor $\hat{\omega}_{11}(k,k) \propto \lambda_1^{2k}$ contributing to the cross-branch correlation decays twice as fast; (ii) if there is a negative eigenvalue, the factor $\hat{\omega}_{11}(k,0) = \hat{\omega}_{11}(0,k) \propto (-1)^k |\lambda_1|^k$ alternates between negative and positive values with an envelope of $|\lambda_1|^k$, while the corresponding contribution to the cross-branch correlation decays monotonically with rate as $|\lambda_1|^{2k}$. Finally, if we have a pair of complex eigenvalues $\lambda_1 = \lambda_2^* = De^{i\Omega}$ then the factors $\hat{\omega}_{i,j}(k,0) = \hat{\omega}_{i,j}(0,k)$ contributing to the lineage correlation function display damped oscillations with frequency $\Omega$ and envelope $D^k$, while the factor $\hat{\omega}_{12}(k,k) = \hat{\omega}_{12}^*(k,k) \propto D^{2k}$ and the factor $\hat{\omega}_{11}(k,k) = \hat{\omega}_{22}^*(k,k) \propto D^{2k} e^{i2\Omega k}$ oscillate with frequency $2\Omega$.

## Determining the period of correlation oscillations from the eigenvalues

We consider the case where the inheritance matrix $\theta$ has a pair of complex conjugate eigenvalues $\lambda^\pm = De^{\pm i2\pi/P}$. The lineage correlation function then oscillates whenever $D \neq \{0,1\}$ and $P \neq \frac{2}{k}, k$ in $\mathbb{Z}$. The period of correlation oscillations per generation is given by

$$\frac{T_0}{\bar{\tau}} = \frac{2\pi}{|\ln(e^{i2\pi/P})|} = \frac{2}{1 - \left| 2(\frac{1}{P} \bmod 1) - 1 \right|}, \tag{M9}$$

where $\mathrm{Arg}(\lambda)$ in $(-\pi, \pi]$ is the argument of the eigenvalue and $\ln(\cdot)$ is the complex logarithm. The former is the angle made between the line joining the origin and the eigenvalue $\lambda$ on the complex plane with the real axis. This means that $T_0/\bar{\tau} = P$ if and only if $P > 2$. Otherwise, $T_0$ is calculated in terms of $P$ by *Equation M9*, (*Appendix 1—figure 8*).

## Data analysis and Bayesian inference of the inheritance matrix model

We determined all pairs of cells in a lineage tree, sorted them by family relations $(k,l)$ and calculated the sample correlation coefficient of interdivision times (*Equation 3*). To maximise the number of samples used to calculate these correlations, an individual cell can appear in more than one pair. For example, if a cell had two cousins, it would be counted in two separate cousin pairs in the cousin-cousin correlation coefficient calculation. For training, we focus on the sample statistics $\hat{X} = (\hat{s}_\tau, \{\hat{\rho}_{(k,l)}\}_{(k,l) \text{ in } C})$ with $C = \{(1,0), (2,0), (1,1), (2,2)\}$ comprised of the interdivision time sample variance and four interdivision time sample correlation coefficients given by the mother-daughter, grandmother-granddaughter, sister-sister and cousin-cousin relations (*Figure 2a*). Note that $\hat{s}_\tau$ is computed across all interdivision times used to calculate the correlation coefficients in each dataset. Errors are estimated using bootstrapping by re-sampling cell pairs with replacement 10,000 times. The resulting variances and correlation coefficients are given in *Appendix 1—table 1*.

The vector of inferred model parameters for the two-dimensional model is $\Theta = (\theta, S_1)$, where we fix $\alpha = (1,1)^\top$ and $S_2 = 0$ for simplicity. A different choice of $\alpha$ did not affect our results (*Appendix 1—figure 4*). Since $S_1$ is symmetric, it consists of the $N$ variances and $N(N-1)/2$ correlation coefficients between the components of $z$. Thus for $N = 2$ the inheritance matrix model has seven free parameters to be estimated. We assumed that the log-likelihood for these statistics is the sum of square errors:

$$-\ln \mathcal{L}(\Theta|\hat{X}) = \frac{(\hat{s}_\tau - s_\tau(\Theta))^2}{\hat{\sigma}_{\hat{s}_\tau}^2} + \sum_{(k,l) \text{ in } C} \frac{(\hat{\rho}_{(k,l)} - \rho_{(k,l)}(\Theta))^2}{\hat{\sigma}_{\hat{\rho}_{(k,l)}}^2}, \tag{M10}$$

which is equivalent to assuming that the sample variance and correlation coefficients are normally distributed for large sample sizes. We calculate the interdivision time variance $s_\tau$ and the generalised tree correlation function $\rho(k, l)$ from *Equation M2* and *Equation M3*. Note that *Equation M2* is the interdivision time variance from a tree where all lineages have the same number of generations, which approximates the variance across all cells in the observed trees (*Appendix 1—table 3*). For simplicity, we neglected possible correlations between the sample statistics in $\hat{X}$ and used bootstrapped estimates for the standard deviation of the sample statistics $\hat{\sigma}_{\hat{s}_\tau}$ and $\hat{\sigma}_{\hat{\rho}(k,l)}$ (*Appendix 1—table 1*). Note that the likelihood is independent of the mean since it is irrelevant for the correlation pattern. We assumed a flat prior with support restricted to $\mathrm{SR}(\boldsymbol{\theta}) < 1$ and $\boldsymbol{S}_1$ positive semi-definite to guarantee the existence of a stationary correlation pattern.

The numerical implementation uses the adaptive Gibbs-sampler implemented in the Julia library `Mamba.jl` (*Smith, 2018*). For each dataset, we sample 11 million parameter sets which include a burn-in transient of 1 million samples. These samples are removed before analysis of the output.

For model comparison we use the AIC (*Akaike, 1974*) given by

$$\mathrm{AIC} = 2k - 2\ln(\hat{\mathcal{L}}), \tag{M11}$$

where $k$ is the number of model parameters and $\ln(\hat{\mathcal{L}})$ is the maximum value of the log-likelihood function given by *Equation M10*. For $\boldsymbol{S}_2 \neq \boldsymbol{0}$, the inheritance matrix model has $k = d(1 + 2d)$ parameters where $d$ is the number of cell cycle factors in the model. For $\boldsymbol{S}_2 = \boldsymbol{0}$ the number of parameters reduces to $k = \frac{1}{2}d(1 + 3d)$.

## Acknowledgements

We thank Bruno Martins, Dimitris Volteras and Paul Piho for their comments on the manuscript. This work has been supported by a scholarship to FAH provided by the EPSRC Centre for Mathematics of Precision Healthcare (EP/N014529/1) and MRC core funding to the London Institute of Medical Sciences (MC-A658-5TY60). ARB is funded by a CRUK Career Development Fellowship (C63833/A25729). PT is funded by a UKRI Future Leaders Fellowship (MR/T018429/1).

## Additional information

### Funding

| Funder | Grant reference number | Author |
|---|---|---|
| EPSRC Centre for Mathematics of Precision Healthcare | EP/N014529/1 | Fern A Hughes |
| MRC London Institute of Medical Sciences | MC-A658-5TY60 | Alexis R Barr |
| UKRI Future Leaders Fellowship | MR/T018429/1 | Philipp Thomas |
| Cancer Research UK | Career Development Fellowship C63833/A25729 | Alexis R Barr |

The funders had no role in study design, data collection and interpretation, or the decision to submit the work for publication.

### Author contributions

Fern A Hughes, Conceptualization, Software, Formal analysis, Validation, Investigation, Visualization, Methodology, Writing – original draft, Writing – review and editing; Alexis R Barr, Philipp Thomas, Conceptualization, Supervision, Funding acquisition, Investigation, Methodology, Writing – original draft, Project administration, Writing – review and editing

### Author ORCIDs

Fern A Hughes  http://orcid.org/0000-0002-4599-5027
Alexis R Barr  http://orcid.org/0000-0002-6684-8114

Philipp Thomas http://orcid.org/0000-0003-4919-8452

**Decision letter and Author response**
Decision letter https://doi.org/10.7554/eLife.80927.sa1
Author response https://doi.org/10.7554/eLife.80927.sa2

## Additional files

### Supplementary files

• Transparent reporting form

### Data availability

The current manuscript is a computational study, so no data have been generated for this manuscript. Modelling code is uploaded to GitHub https://github.com/fernhughes/Lineage-tree-correlation-pattern-inference (copy archived at swh:1:rev:dc69bbce5ce909813d7d4356c9fd2da045e02c79).

The following previously published dataset was used:

| Author(s) | Year | Dataset title | Dataset URL | Database and Identifier |
| --- | --- | --- | --- | --- |
| Martins BMC, Tooke AK, Thomas P, Lock JCW | 2018 | Research data supporting Cell size control driven by the circadian clock and environment in cyanobacteria | https://doi.org/10.17863/CAM.31834 | Apollo - University of Cambridge Repository, 10.17863/CAM.31834 |

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

## Appendix 1

### A1. Small noise approximation

Here, we will derive the inheritance matrix model given by *Equation 2a*, and *b* in the main text. We assume that the fluctuations in the hidden cell factor dynamics are small, which leads to a computationally efficient approximation.

Firstly, in the limit of zero fluctuations, all cells must be identical. Hence, all cell cycle factors are equal to their means $\boldsymbol{\mu} = (\mu_1, \mu_2, \cdots, \mu_N)^\top = \mathbf{E}(\boldsymbol{y}_p)$ and similarly for the noise vectors $\beta = (\beta_1, \beta_2, \cdots, \beta_N)^\top = \mathbf{E}(e_{2m}) = \mathbf{E}(e_{2m+1})$ in *Equation 1b*. From *Equation 1a*, and *b* we then find that

$$\bar{\tau} = f(\boldsymbol{\mu}), \ \boldsymbol{\mu} = \boldsymbol{g}(\boldsymbol{\mu}) + \boldsymbol{\beta} \tag{A1}$$

which can be efficiently solved for $\bar{\tau}$ and $\mu$ using standard numerical methods.

Secondly, we can decompose the interdivision time and the cell cycle factor vector into their respective mean and fluctuating components by

$$\tau = \hat{\tau} + \tau_p', \ \boldsymbol{y}_p = \boldsymbol{\mu} + \bar{x}_p. \tag{A2}$$

Denoting the index of the present cell by $p$ and the one of its mother by $m$, we can expand $f$ and $g$ around the limit of zero fluctuations and we obtain to leading order

$$\begin{aligned} f(\boldsymbol{y}_p) &= f(\boldsymbol{\mu}) + \alpha^\top(\boldsymbol{y}_p - \boldsymbol{\mu}) + \cdots \\ g(\boldsymbol{y}_m) &= g(\boldsymbol{\mu}) + \theta(\boldsymbol{y}_m - \boldsymbol{\mu}) + \cdots \end{aligned} \tag{A3}$$

$$\tilde{\alpha}_i = \left.\frac{\partial f(\boldsymbol{y})}{\partial y_i}\right|_{\boldsymbol{y}=\boldsymbol{\mu}}, \qquad \tilde{\theta}_i j = \left.\frac{\partial g_i(\boldsymbol{y})}{\partial y_j}\right|_{\boldsymbol{y}=\boldsymbol{\mu}}. \tag{A4}$$

where

$$\begin{aligned} \bar{\tau} + \tau_p' &= f(\boldsymbol{\mu}) + \boldsymbol{\alpha}^\top \tilde{\boldsymbol{x}}_p + \cdots, \\ \boldsymbol{\mu} + \tilde{\boldsymbol{x}}_p &= \left(g(\boldsymbol{\mu}) + \boldsymbol{\theta}\tilde{\boldsymbol{x}}_m + \cdots\right) + \boldsymbol{\beta} + \tilde{z}_p, \end{aligned} \tag{A5}$$

Using this expansion and *Equation A2* in *Equation 1a* and *b* of the main text we arrive at

$$\bar{\tau} + \tau_p' \quad = f(\boldsymbol{\mu}) + \tilde{\boldsymbol{\alpha}}^\top \tilde{\boldsymbol{x}}_p + \cdots, \tag{A6}$$

$$\boldsymbol{\mu} + \tilde{\boldsymbol{x}}_p \quad = \left(\boldsymbol{g}(\boldsymbol{\mu}) + \tilde{\boldsymbol{\theta}}\tilde{\boldsymbol{x}}_m + \ldots\right) + \boldsymbol{\beta} + \tilde{z}_p, \tag{A7}$$

where we have set $\boldsymbol{e}_p = \boldsymbol{\beta} + \tilde{z}_p$ and $\tilde{z}_p = (\tilde{z}_{p,1}, \tilde{z}_{p,2}, \cdots, \tilde{z}_{p,N})^\top$ giving the fluctuations around the mean for the noise vectors. Comparing *Equation A7* with *Equation A1* and collecting terms to leading order, we obtain the linearised system:

$$\tau_p' = \tilde{\boldsymbol{\alpha}}^\top \tilde{\boldsymbol{x}}_p \tag{A8}$$

$$\tilde{\boldsymbol{x}}_p = \tilde{\boldsymbol{\theta}}\tilde{\boldsymbol{x}}_m + \tilde{z}_p. \tag{A9}$$

Next, we define the diagonal scaling matrix $\boldsymbol{\Gamma}$ with non-zero elements as

$$\Gamma_{ii} = \begin{cases} 1 & \text{if} \quad \tilde{\alpha}_i = 0, \\ \tilde{\alpha}_i & \text{otherwise,} \end{cases}, \tag{A10}$$

for $i = 1, 2, \ldots, N$. Using the rescaled noise sources $z = \boldsymbol{\Gamma}\tilde{z}$, we find the rescaled inheritance matrix $\boldsymbol{\theta}$ and $\boldsymbol{\alpha}$-coefficients

$$\alpha_i = \begin{cases} 0 & \text{if} \quad \tilde{\alpha}_i = 0, \\ 1 & \text{otherwise,} \end{cases} \qquad i = 1, 2, \ldots, N, \tag{A11}$$

$$\boldsymbol{\theta} = \boldsymbol{\Gamma}\tilde{\boldsymbol{\theta}}\boldsymbol{\Gamma}^{-1}. \tag{A12}$$

The rescaled cell cycle factor fluctuations $x_m = \Gamma\tilde{x}_m$ follow *Equation 2* of the main text and we reach rescaled variance-covariance matrices $S_1$ and $S_2$ as follows

$$S_1 = \boldsymbol{\Gamma}\tilde{S}_1\boldsymbol{\Gamma}^{\top}, \quad S_2 = \boldsymbol{\Gamma}\tilde{S}_2\boldsymbol{\Gamma}^{\top}. \tag{A13}$$

## A2. Beyond the small noise approximation: cell cycle factor complexes account for nonlinear fluctuations

Here we analyse the effect of nonlinearity on the interdivision time correlation patterns. For simplicity we consider a single cell cycle factor and follow the same lines as in Appendix 1 - Section A1, *Equation A9*, while including terms of order $x^2$. This leads to the expansion interdivision time and factor fluctuations

$$\tau_p' = \tilde{\alpha}^{\top}\tilde{x}_p + \tilde{\beta}^{\top}\tilde{x}_p^2 + O(x^3), \tag{A14}$$

$$\tilde{x}_p = \tilde{\theta}\tilde{x}_m + \tilde{H}\tilde{x}_m^2 + \tilde{z}_p + O(x^3), \tag{A15}$$

where $\tilde{\theta}$ is the Jacobian of the cell cycle factor dynamics, as before, and $\tilde{H} = g''(\mu_x)$ is the Hessian. From the second equation we obtain

$$\tilde{x}_p^2 = \tilde{\theta}^2\tilde{x}_m^2 + 2\tilde{\theta}\tilde{z}_p\tilde{x}_m + \tilde{z}_p^2 + O(x^3). \tag{A16}$$

Defining $\tilde{X}_p = (\tilde{x}_p, \tilde{x}_p^2)$ and $\tilde{Z}_p = (\tilde{z}_p, \tilde{z}_p^2)$, combining *Equation A14* and *Equation A16* and rescaling variables as in *Equation A12* and *Equation A13*, we find the extended inheritance matrix model

$$\begin{aligned} \tau &= \mu + A^{\top}X_p, \\ X_{2m} &= \Theta X_m + B(X_m)Z_{2m}, \\ X_{2m+1} &= \Theta X_m + B(X_m)Z_{2m+1} \end{aligned} \tag{A17}$$

where

$$\Theta = \begin{pmatrix} \theta & H \\ 0 & \theta^2 \end{pmatrix}, \quad B(X_m) = \begin{pmatrix} 1 & 0 \\ 2x_m & 1 \end{pmatrix}, \quad A = \begin{pmatrix} \alpha \\ \beta \end{pmatrix}. \tag{A18}$$

Here $H = \tilde{H}\tilde{\alpha}/\tilde{\beta}$ and $\beta = 1$ if $\tilde{\beta} \neq 0$, and analogously, $H = \tilde{H}\tilde{\alpha}$ and $\beta = 0$ if $\tilde{\beta} = 0$. Hence, the interdivision time correlation patterns with small to moderate fluctuations can be described through an extended linear system (*Equation A7*) that includes nonlinear terms $x_p^2$. These additional terms can be interpreted as cell cycle factors forming binary complexes. The presence of these complexes increases the number of cell cycle factors and extends the eigenvalue spectrum of the effective inheritance matrix $\Theta$ by $\theta^2$. Hence, the presence of complexes leads to mixed correlation patterns. For example, for a single cell cycle factor, the eigenvalues of $\Theta$ are $(\theta, \theta^2)$, which corresponds to an alternator pattern for $\theta < 0$. More generally, we may expect that nonlinear patterns can be described through mixtures of aperiodic, alternator, and oscillatory patterns. For example, the complex eigenvalue spectrum of an oscillator pattern ($e^{\pm i2\pi/P}$) will include powers of complex eigenvalues ($e^{\pm i4\pi/P}$) resulting in harmonics of the fundamental correlation oscillation frequency similar to higher order harmonics observed in single-cell time-series of the circadian clock (*Martins et al., 2016*; *Thomas et al., 2013*).

## A3. Derivation of the generalised tree correlation function

In this section we derive an analytical expression for the *generalised tree correlation function*. This gives the Pearson correlation coefficient in interdivision time for any pair of related cells. We start with the equation for the Pearson correlation coefficient, and from there derive a formula for the

interdivision time covariance using the known properties of the cell cycle factors $\boldsymbol{x}$. From this, we can derive the general formula for the correlation coefficient between any related cell pair.

We associate a cell pair with an index $(k, l)$ which measures the distance to the nearest common ancestor as given in 'The inheritance matrix model reveals threedistinct interdivision time correlation patterns' (*Figure 2a*). From this, we denote their interdivision time fluctuations as $\tau'_k$ and $\tau'_l$ respectively. The Pearson correlation coefficient between these fluctuations is given by

$$\rho(k, l) = \frac{\mathrm{Cov}(\tau'_k, \tau'_l)}{s_{\tau'}}, \tag{A19}$$

where $s_{\tau'}$ is the variance of the interdivision time fluctuations.

The interdivision time fluctuations $\tau'_k$ and $\tau'_l$ are calculated from the vector of rescaled cell cycle factor fluctuations $\boldsymbol{x}_k$ as given in 'A general inheritance matrix model provides a unified framework for lineage tree correlation patterns', giving the equations

$$\tau'_k = \boldsymbol{\alpha}^\top \boldsymbol{x}_k \quad \text{and} \quad \tau'_l = \boldsymbol{\alpha}^\top \boldsymbol{x}_l. \tag{A20}$$

Substituting *Equation A20* into *Equation A19*, we obtain a formula for in terms of the cell cycle factor fluctuations and the coefficients alone

$$\rho(k, l) \quad = \frac{\mathrm{Cov}(\boldsymbol{\alpha}^\top \boldsymbol{x}_k, \boldsymbol{\alpha}^\top \boldsymbol{x}_l)}{\sqrt{\mathrm{Var}(\boldsymbol{\alpha}^\top \boldsymbol{x}_k)}\sqrt{\mathrm{Var}(\boldsymbol{\alpha}^\top \boldsymbol{x}_l)}}, \tag{A21}$$

$$= \frac{\boldsymbol{\alpha}^\top \mathrm{Cov}(\boldsymbol{x}_k, \boldsymbol{x}_l)\boldsymbol{\alpha}}{\sqrt{\boldsymbol{\alpha}^\top \mathrm{Var}(\boldsymbol{x}_k)\boldsymbol{\alpha}}\sqrt{\boldsymbol{\alpha}^\top \mathrm{Var}(\boldsymbol{x}_l)\boldsymbol{\alpha}}}. \tag{A22}$$

Since $\boldsymbol{x}_k$ and $\boldsymbol{x}_l$ are identically distributed in steady state, we have that $\mathrm{Var}(\boldsymbol{x}_k) = \mathrm{Var}(\boldsymbol{x}_l) = \mathrm{Cov}(\boldsymbol{x}, \boldsymbol{x}) = \boldsymbol{\Sigma}$ as specified in Materials and methods - 'Analytical solution of the inheritance matrix model''. We can write $\rho(k, l)$ now as

$$\rho(k, l) = \frac{\boldsymbol{\alpha}^\top \mathrm{Cov}(\boldsymbol{x}_k, \boldsymbol{x}_l)\boldsymbol{\alpha}}{\boldsymbol{\alpha}^\top \boldsymbol{\Sigma} \boldsymbol{\alpha}}, \tag{A23}$$

where $\boldsymbol{\alpha}^\top \boldsymbol{\Sigma} \boldsymbol{\alpha}$ gives the variance of the interdivision time fluctuations $\tau'$.

Using the model *Equation 2* we can write the formula for the $\boldsymbol{x}$ vectors for the two cells in the cell pair $(k, l)$ as

$$\boldsymbol{x}_k \quad = \boldsymbol{\theta} \boldsymbol{x}_{k-1} + \boldsymbol{z}_k, \tag{A24}$$

$$\boldsymbol{x}_l \quad = \boldsymbol{\theta} \boldsymbol{x}_{l-1} + \boldsymbol{z}_l, \tag{A25}$$

where cells $k$ and $l$ have mother cells $k - 1$ and $l - 1$ respectively. The two cells are sisters if and only if their subscripts are both equal to 1, meaning they share a mother cell. Using recurrence of the model, we can write these equations as

$$\begin{aligned} \boldsymbol{x}_k &= \boldsymbol{\theta}^k \boldsymbol{x}_0 + \sum_{i=1}^{k} \boldsymbol{\theta}^{k-i} \boldsymbol{z}_i, \\ \boldsymbol{x}_l &= \boldsymbol{\theta}^l \boldsymbol{x}_0 + \sum_{j=1}^{l} \boldsymbol{\theta}^{l-j} \boldsymbol{z}_j, \end{aligned} \tag{A26}$$

where $\boldsymbol{x}_0$ is the vector of cell cycle factors for the most recent common ancestor for a cell pair given by $(k, l)$.

All that remains is to derive a function for $\mathrm{Cov}(\boldsymbol{x}_k, \boldsymbol{x}_l)$ which we will denote $\boldsymbol{\omega}(k, l)$. We calculate $\boldsymbol{\omega}(k, l)$ as follows using expectations:

$$\boldsymbol{\omega}(k, l) = \mathrm{Cov}(\boldsymbol{x}_k, \boldsymbol{x}_l) \quad = \mathrm{E}\left[\left(\boldsymbol{x}_k - \boldsymbol{\mu}_{\boldsymbol{x}_k}\right)\left(\boldsymbol{x}_l - \boldsymbol{\mu}_{\boldsymbol{x}_l}\right)^\top\right], \tag{A27}$$

$$= \mathrm{E}\left[\boldsymbol{x}_k \boldsymbol{x}_l^\top\right] - \boldsymbol{\mu}_{\boldsymbol{x}_k}\boldsymbol{\mu}_{\boldsymbol{x}_l}^\top, \tag{A28}$$

where $\boldsymbol{\mu}_{\boldsymbol{x}_k}$ and $\boldsymbol{\mu}_{\boldsymbol{x}_l}$ are the mean vectors of $\boldsymbol{x}_k$ and $\boldsymbol{x}_l$ respectively which are both equal to 0, giving

$$\boldsymbol{\omega}(k, l) = \mathrm{E}\left[\boldsymbol{x}_k \boldsymbol{x}_l^\top\right]. \tag{A29}$$

To find $\mathrm{E}\left[\boldsymbol{x}_k \boldsymbol{x}_l^\top\right]$ in terms of the model parameters, we substitute in *Equation A26* for $\boldsymbol{x}_k$ and $\boldsymbol{x}_l$ and get

$$\mathrm{E}\left[\boldsymbol{x}_k \boldsymbol{x}_l^\top\right] = \mathrm{E}\left[\left(\boldsymbol{\theta}^k \boldsymbol{x}_0\right)\left(\boldsymbol{\theta}^l \boldsymbol{x}_0\right)^\top + \left(\sum_{i=1}^{k} \boldsymbol{\theta}^{k-i} \boldsymbol{z}_k\right)\left(\sum_{j=1}^{l} \boldsymbol{\theta}^{l-j} \boldsymbol{z}_j\right)^\top\right].\tag{A30}$$

The noise term fluctuations $z$ are only correlated if the cells are sisters, which only occurs when the distance we have $(k, l) = (1, 1)$. So for the summations above, we exclude all terms except where $i = j = 1$. Doing this and expanding we get

$$\mathrm{E}\left[\boldsymbol{x}_k \boldsymbol{x}_l^\top\right] = \boldsymbol{\theta}^k \mathrm{E}\left[\boldsymbol{x}_0 \boldsymbol{x}_0^\top\right]\left(\boldsymbol{\theta}^l\right)^\top + \delta_{k \geq 1}\delta_{l \geq 1}\boldsymbol{\theta}^{k-1}\mathrm{E}\left[\boldsymbol{z}_1 \boldsymbol{z}_1^\top\right]\left(\boldsymbol{\theta}^{l-1}\right)^\top,\tag{A31}$$

where $\delta_{k \geq 1}$ and $\delta_{l \geq 1}$ are given in *Equation M4*. We also have that

$$\mathrm{Cov}(\boldsymbol{x}_0, \boldsymbol{x}_0) = \mathrm{E}\left[\boldsymbol{x}_0 \boldsymbol{x}_0^\top\right].\tag{A32}$$

The matrix $\mathrm{Cov}(\boldsymbol{x}_0, \boldsymbol{x}_0)$ is equivalent to the covariance matrix for any $\boldsymbol{x}$, giving $\mathrm{Cov}(\boldsymbol{x}_0, \boldsymbol{x}_0) = \boldsymbol{\Sigma}$. This gives

$$\mathrm{E}\left[\boldsymbol{x}_0 \boldsymbol{x}_0^\top\right] = \mathrm{Cov}(\boldsymbol{x_0}, \boldsymbol{x_0}),\tag{A33}$$

$$= \boldsymbol{\Sigma}.\tag{A34}$$

Similarly we have,

$$\mathrm{Cov}(\boldsymbol{z}_{2m}, \boldsymbol{z}_{2m+1}) = \mathrm{E}\left[\boldsymbol{z}_1 \boldsymbol{z}_1^\top\right].\tag{A35}$$

As $\mathrm{E}(\boldsymbol{z}) = \boldsymbol{0}$, and $\mathrm{Cov}(\boldsymbol{z}_{2m}, \boldsymbol{z}_{2m+1}) = \boldsymbol{S}_2$ as stated in Materials and methods - 'Analytical solution of the inheritance matrix model', we obtain,

$$\mathrm{E}\left[\boldsymbol{z}_1 \boldsymbol{z}_1^\top\right] = \boldsymbol{S}_2.\tag{A36}$$

*Equation A31* therefore becomes:

$$\mathrm{E}\left[\boldsymbol{x}_k \boldsymbol{x}_l^\top\right] = \boldsymbol{\theta}^k \boldsymbol{\Sigma}\left(\boldsymbol{\theta}^l\right)^\top + \delta_{k \geq 1}\delta_{l \geq 1}\boldsymbol{\theta}^{k-1}\boldsymbol{S}_2\left(\boldsymbol{\theta}^{l-1}\right)^\top.\tag{A37}$$

Substituting *Equation A37* back into *Equation A29* we get

$$\boldsymbol{\omega}(k, l) = \boldsymbol{\theta}^k \boldsymbol{\Sigma}\left(\boldsymbol{\theta}^l\right)^\top + \delta_{k \geq 1}\delta_{l \geq 1}\boldsymbol{\theta}^{k-1}\boldsymbol{S}_2\left(\boldsymbol{\theta}^{l-1}\right)^\top,\tag{A38}$$

giving us the final equation for $\boldsymbol{\omega}(k, l)$. Using the above equation in *Equation A23*, we obtain *Equation M3* of the Methods.

## A4. Derivation of the formula for the oscillator periods, $T_n$

The period of correlation oscillation as observed in the lineage correlation functions is given by *Thomas et al., 2018*. We can reveal the underlying oscillator periods by shifting the inferred period $T_0$ to obtain a smaller period $T_n$. This means that shorter periods would produce the same inferred period in the lineage correlation function when sampled at the original frequency of once per cell cycle (*Figure 5a*).

The oscillator periods are obtained by adding or subtracting multiples of $2\pi$ to the argument of the eigenvalue which results in the new argument being in the same position in the complex plane. The oscillator period $T_n$ with shift $n$ in $\mathbb{Z}$ is therefore given by

$$T_n = \bar{\tau}\frac{2\pi}{|\mathrm{Arg}(\lambda) + 2\pi n|}.\tag{A39}$$

Taking *Equation A40* and substituting in *Equation 6*, we obtain $T_n$ in terms of $T_0$ as *Equation 7*.

## A5. Solution of the tree correlation function and parameter identifiability for simple inheritance rules

We consider the limiting case of a single cell cycle factor ($N = 1$) resulting in simple inheritance rules. This situation could model a growth factor that can either increase or decrease interdivision times of cells depending on the monotonicity of $f$ in *Equation 1*. The analytical solution (*Equation M7*) of the inheritance matrix model then reduces to

$$\rho(k,l) = w\theta^{k+l}, \quad s_\tau = \frac{S_1}{1-\theta^2},$$ (A40)

where $w = 1 + \delta_{k\geq1}\delta_{l\geq1}\frac{S_2}{S_1}\frac{(1-\theta^2)}{\theta^2}$. First, we observe that, given single cell measurements of the mother-daughter correlation coefficient $\rho(0,1)$, the daughter-daughter correlation coefficient $\rho(1,1)$ and the variance $s_\tau$, the parameters $\theta$, $S_1$ and $S_2$ are uniquely identifiable:

$$\theta = \rho(1,0), \quad S_1 = s_\tau(1-\rho^2(1,0)), \quad S_2 = \frac{\rho(1,1)-\rho^2(1,0)}{1-\rho^2(1,0)}.$$ (A41)

Thus measurements of the variance, lineage- and cross-branch correlations fully determine the parameters. The tree correlation function is, however, independent of $f$, which means that the interdivision time correlation pattern carries no information whether the growth factor increases or decreases growth. The reason for this indifference is that cell cycle factors are identified only by their fluctuation pattern, i.e., for each cell cycle factor whose fluctuations increase interdivision time $x$, we could define another cell cycle factor fluctuation that decrease interdivision time $-x$. We accounted for this unidentifiability issue trough a similarity transformation using the scaling matrix $\boldsymbol{\Gamma}$ in *Equation A12* and *Equation A13* that transforms all cell cycle factor fluctuations to increase interdivision time. Of course, this unidentifiabiliy could be removed through explicitly measuring the involved cell cycle factors.

## A6. Mapping mechanistic cell cycle and cell size control models to the inheritance matrix model

To further investigate the output of the inheritance matrix model, we propose multiple models of known cell cycle control mechanisms, and map them to our inheritance matrix model framework. All cell size models assume symmetric division.

### A6.1 Cell size control model with correlated growth

Considering the influence of cell size control on interdivision time (*Taheri-Araghi et al., 2015*; *Kohram et al., 2021*; *Ho et al., 2018*), here we propose a cell size control model where we have some mother to daughter inheritance of both the added size $\Delta$ and the growth rate $\kappa$ (*Appendix 1—figure 5g*). The model equations are given by:

$$\begin{aligned} s_{b,2m} &= \frac{1}{2}\left(as_{b,m} + \Delta_m\right), & \Delta_{2m} &= b\Delta_m + \xi_{2m}, & \kappa_{2m} &= c\kappa_m + \phi_{2m}, \\ s_{b,2m+1} &= \frac{1}{2}\left(as_{b,m} + \Delta_m\right), & \Delta_{2m+1} &= b\Delta_m + \xi_{2m+1}, & \kappa_{2m+1} &= c\kappa_m + \phi_{2m+1}. \end{aligned}$$ (A42)

The noise terms $\xi$ and $\phi$ are independent between sisters such that $\text{Cov}(\xi_{2m},\xi_{2m+1}) = \text{Cov}(\phi_{2m},\phi_{2m+1}) = 0$. Assuming exponential growth the formula for the interdivision time is given by

$$\tau_p = \frac{\ln\left|a+\frac{\Delta_p}{s_{b,p}}\right|}{\kappa_p},$$ (A43)

where $p$ represents the index of a given cell. Taking the vector of cell cycle factors for the mother cell to be $\boldsymbol{y}_m = (y_{m,1}, y_{m,2}, y_{m,3})^\top = (\Delta_m, s_{b,m}, \kappa_m)^\top$ and comparing *Equation 1a* and *b* with *Equation A43* and *Equation A44*, we obtain

$$f(\boldsymbol{y}) = \frac{\ln\left|a+\frac{y_1}{y_2}\right|}{y_3}, \quad \boldsymbol{g}(\boldsymbol{y}) = \left(by_1, \frac{1}{2}\left(ay_2 + y_1\right), cy_3\right)^\top, \quad \boldsymbol{\beta} = (\text{E}[\xi], 0, \text{E}[\phi])^\top.$$ (A44)

Then we can calculate the means from *Equation A1*,

$$\mu_1 = \frac{\text{E}[\xi]}{b-1}, \quad \mu_2 = \frac{\text{E}[\xi]}{(a-2)(b-1)}, \quad \mu_3 = -\frac{\text{E}[\phi]}{c-1}.$$ (A45)

Then using *Equation A45* in *Equation A5* and *Equation A12*, we find

$$\boldsymbol{\alpha} = \begin{pmatrix} 1 \\ 1 \\ 1 \end{pmatrix}, \quad \boldsymbol{\theta} = \begin{pmatrix} b & 0 & 0 \\ \frac{1}{2}(a-2) & \frac{a}{2} & 0 \\ 0 & 0 & c \end{pmatrix}. \tag{A46}$$

Assuming $\tilde{S}_2 = \mathbf{0}$ and using *Equation A13*, we find

$$S_1 = \begin{pmatrix} \frac{(a-2)^2(b-1)^2(c-1)^2\mathrm{Var}[\xi_{2m}]}{4\mathbf{E}[\xi]^2\mathbf{E}[\phi]^2} & 0 & \frac{(a-2)(b-1)(c-1)^3\ln|2|\mathrm{Var}[\phi_{2m}]\mathrm{Var}[\xi_{2m}]}{2\mathbf{E}[\xi]\mathbf{E}[\phi]^3} \\ 0 & 0 & 0 \\ \frac{(a-2)(b-1)(c-1)^3\ln|2|\ \mathrm{Var}[\phi_{2m}]\mathrm{Var}[\xi_{2m}]}{2\mathbf{E}[\xi]\mathbf{E}[\phi]^3} & 0 & \frac{(c-1)^4\ln|2|^2\ \mathrm{Var}[\phi_{2m}^2]}{\mathbf{E}[\phi]^4} \end{pmatrix}. \tag{A47}$$

The $\boldsymbol{\theta}$ matrix has eigenvalues $\boldsymbol{\lambda} = (\frac{a}{2}, b, c)$ which give an aperiodic pattern for $a, b, c > 0$ and an alternator pattern otherwise (*Appendix 1—figure 5i and j*). These same patterns arise for all real eigenvalues in the 3D model in the same was as in the two-dimensional system. Only a single negative eigenvalue is needed for the lineage correlation function to display an alternator pattern. We are restricted to $a$ in $(-2, 2)$ and $b, c$ in $(-1, 1)$ to ensure $\mathrm{SR}(\boldsymbol{\theta}) < 1$. The cousin-mother inequality for this system is too complex to be looked at analytically, so we use numerical methods to visualise the parameter region in which the cousin-mother inequality can be satisfied (*Appendix 1—figure 5h*).

For the case of the aperiodic pattern, we observe positive same factor mother-daughter correlation and negative alternate factor mother-daughter correlation (*Appendix 1—figure 5k*). In contrast, for an alternator pattern, the mother daughter same factor correlation is negative, but the alternate factor correlations vary between positive and negative values (*Appendix 1—figure 5l*).

## A6.2 Simple cell size control model

For the special case of $b = \mathrm{Var}[\phi] = 0$ and $c = 1$, the model reduces to a simple cell size control model with fluctuating added size (*Appendix 1—figure 5a*). The inheritance matrix $\boldsymbol{\theta}$ then has eigenvalues $\boldsymbol{\lambda} = (0, \frac{a}{2})$. Thus depending on the choice of $a$, this model can produce both an alternator and aperiodic pattern (*Appendix 1—figure 5c and d*). In this case, using *Equation M3* the cousin-mother inequality becomes

$$a^2(a-2) + 4|a-2| < 0, \tag{A48}$$

which cannot be satisfied for $|a| < 2$, which implies $\mathrm{SR}(\boldsymbol{\theta}) = \frac{a}{2} < 1$. Hence the cousin-mother inequality cannot be satisfied for any reasonable choice of $a$ in this simple model (*Appendix 1—figure 5b*).

For an aperiodic pattern, this simplified model exhibits positive same factor mother-daughter correlation and negative alternate factor mother-daughter correlation (*Appendix 1—figure 5e*). In the alternator case, this model exhibits negative same factor mother-daughter correlation and also negative alternate factor mother-daughter correlation (*Appendix 1—figure 5f*).

## A6.3 Abstract cell cycle phase model

We propose a model of two abstract cell cycle phases that have no integrated dependence on cell size (*Appendix 1—figure 5m*). The model equations are given by

$$\begin{aligned} y_{2m,1} &= ay_{m,1} + by_{m,2} + \xi_{2m}, && \text{and} & y_{2m,2} &= cy_{m,2} + \phi_{2m}, \\ y_{2m+1,1} &= ay_{m,1} + by_{m,2} + \xi_{2m+1}, && \text{and} & y_{2m+1,2} &= cy_{m,2} + \phi_{2m+1}. \end{aligned} \tag{A49}$$

The noise terms $\xi$ and $\phi$ are independent between sister cells such that $\mathrm{Cov}(\xi_{2m}, \xi_{2m+1}) = \mathrm{Cov}(\phi_{2m}, \phi_{2m+1}) = 0$. In this case we have that the two factors make up the length of the cell cycle, so we simply have $\tau_p = y_{p,1} + y_{p,2}$.

Therefore using *Equation 1a* and *b*, we obtain

$$f(\mathbf{y}) = y_1 + y_2, \quad \mathbf{g}(\mathbf{y}) = (ay_1 + by_2, cy_2)^\top, \quad \boldsymbol{\beta} = (\mathbf{E}[\xi], \mathbf{E}[\phi])^\top \tag{A50}$$

We calculate the means from *Equation A1*,

$$\mu_1 = \frac{b\mathbf{E}[\phi]+(1-c)\mathbf{E}[\xi]}{(1-a)(1-c)}, \quad \text{and} \quad \mu_2 = \frac{\mathbf{E}[\phi]}{1-c}. \tag{A51}$$

Then using *Equation A51* in *Equation A5* and *Equation A12*, we find

$$\boldsymbol{\alpha}=\begin{pmatrix} 1 \\ 1 \end{pmatrix}, \quad \boldsymbol{\theta}=\begin{pmatrix} a & b \\ 0 & c \end{pmatrix}. \tag{A52}$$

As the noise terms are independent between sisters we have $\tilde{S}_2 = \mathbf{0}$ and using *Equation A13* we obtain

$$S_1 = \begin{pmatrix} \text{Var}[\xi_{2m}^2] & \text{Var}[\xi_{2m}]\text{Var}[\phi_{2m}] \\ \text{Var}[\xi_{2m}]\text{Var}[\phi_{2m}] & \text{Var}[\phi_{2m}^2] \end{pmatrix}. \tag{A53}$$

The inheritance matrix $\boldsymbol{\theta}$ has eigenvalues $\boldsymbol{\lambda} = (a, c)$ which gives an aperiodic pattern for $a$ and $c > 0$ and an alternator pattern otherwise (*Appendix 1—figure 5o, p*).

The analytical form of the cousin-mother inequality is complex so we use numerical methods to visualise the parameter region in which the cousin-mother inequality can be satisfied (*Appendix 1—figure 5n*).

We calculate individual factor mother-daughter correlations and find that for an aperiodic pattern, the model exhibits a range of correlation patterns (*Appendix 1—figure 5q*). However, for an alternator pattern, we obtain positive same factor mother-daughter correlation and negative alternate factor mother-daughter correlation (*Appendix 1—figure 5r*)

## A7. Models of circadian-clock-driven correlation patterns

### A7.1 Kicked cell cycle model

Here we analyse the kicked cell cycle model (*Mosheiff et al., 2018*) with our framework (*Appendix 1—figure 6a*). We will propose an inheritance matrix and then show that it reduces to the kicked cell cycle model for certain parameter choices. Consider the $3 \times 3$ inheritance matrix $\boldsymbol{\theta}$ and noise vector $z_n$ given by

$$\boldsymbol{\theta} = \begin{pmatrix} \beta & 1 & 1 \\ 0 & D\cos\frac{2\pi}{P} & D\sin\frac{2\pi}{P} \\ 0 & -D\sin\frac{2\pi}{P} & D\cos\frac{2\pi}{P} \end{pmatrix} \quad \text{and} \quad z_n = \begin{pmatrix} \xi_{n,\tau} \\ \xi_{n,1} \\ \xi_{n,2} \end{pmatrix}, \tag{A54}$$

for $n \in \{2m, 2m + 1\}$. We have that $S_1$ is given by $\text{Cov}(z_{2m}, z_{2m})$, however we assume that the noise terms $\xi_n$ are independent between sisters such that $S_2 = \text{Cov}(z_{2m}, z_{2m+1}) = \mathbf{0}$. Assuming $\boldsymbol{\alpha} = (1, 0, 0)^\top$, the interdivision times are governed by

$$\begin{aligned} \tau_{2m} &= \beta\tau_m + \hat{x}_{m,1} + \hat{x}_{m,2} + z_{2m}, \\ \tau_{2m+1} &= \beta\tau_m + \hat{x}_{m,1} + \hat{x}_{m,2} + z_{2m+1}. \end{aligned} \tag{A55}$$

The oscillator is represented by the cell cycle factors $\hat{x}$ that evolve according to

$$\begin{aligned} \hat{x}_{2m} &= \hat{\boldsymbol{\theta}}\hat{x}_m + \hat{z}_{2m}, \\ \hat{x}_{2m+1} &= \hat{\boldsymbol{\theta}}\hat{x}_m + \hat{z}_{2m+1}, \end{aligned} \tag{A56}$$

with oscillator inheritance matrix

$$\hat{\boldsymbol{\theta}}=\begin{pmatrix} D\cos\frac{2\pi}{P} & D\sin\frac{2\pi}{P} \\ -D\sin\frac{2\pi}{P} & D\cos\frac{2\pi}{P} \end{pmatrix}. \tag{A57}$$

We can solve *Equation A56* along an ancestral lineage of $n$ generations

$$\hat{x}_n = \hat{\theta}^n \hat{x}_0 + \sum_{i=1}^{n} \hat{\theta}^{n-i} \hat{z}_i, \tag{A58}$$

where $\hat{x}_0$ is the state of the ancestral cell. Substituting *Equation A58* into *Equation A55* and assuming $\hat{z}_i = 0$, i.e., the cell cycle oscillator $\hat{x}$ is deterministic, the interdivision time of the mother determines the interdivision time of the daughter cell via

$$\tau_n = \beta\tau_{n-1} + D^n \left( \hat{x}_0^+ \cos\frac{2\pi(n-1)}{P} + \hat{x}_0^- \sin\frac{2\pi(n-1)}{P} \right) + z_n, \tag{A59}$$

where $\hat{x}_0^+ = (\hat{x}_{0,1} + \hat{x}_{0,2})$ and $\hat{x}_0^- = (\hat{x}_{0,1} - \hat{x}_{0,2})$ which represent initial conditions. Assuming $t_n = \sum_{i=1}^{n} \tau_n \approx n\bar{\tau}$ approximates the time at birth for $n \gg 1$, this leads to

$$\tau_n = \beta\tau_{n-1} + D^n \left( \hat{x}_0^+ \cos\frac{2\pi t_n \bar{\tau}}{P} + \hat{x}_0^- \sin\frac{2\pi t_n \bar{\tau}}{P} \right) + z_n, \tag{A60}$$

Comparing *Equation A60* to Equations 1 and 2 in *Mosheiff et al., 2018*, we see that our IMM agrees with the kicked cell cycle model when $D = 1$, $\hat{x}_0^+ = 0$, $\xi_{n,1} = \xi_{n,2} = 0$, and large $n$.

## A7.2 Circadian-clock-driven cell size control model

Here we analyse the model of cell size control driven by the circadian clock proposed in *Martins et al., 2018* within the inheritance matrix model framework (*Appendix 1—figure 6e*). The division rate, $\Gamma(s, s_b, \frac{\partial s}{\partial t}, t)$ in Equation 1 of *Martins et al., 2018* is given by

$$\Gamma(s, s_b, \tfrac{\partial s}{\partial t}, t) = G(t)S(s, s_b)\tfrac{\partial s}{\partial t} \tag{A61}$$

where $s$ is the cell size with $s_b$ being the size at birth. $G(t)$ is a function of time $t$ that couples the size control to the circadian clock, and $S(s, s_b)$ is the division rate per unit volume of the cell. Assuming cells grow exponentially with growth rate $\alpha$, we have

$$s(\tau) = s_b e^{\alpha\tau} \quad \text{and} \quad t(s, t_b) = t_b + \frac{1}{\alpha} \ln\frac{s}{s_b}, \tag{A62}$$

and the division size follows

$$P(s_d|t_b, s_b) = G(t(s_d, t_b))S(s_d, s_b) \exp[-\int_{s_b}^{s_d} ds\, G(t(s, t_b))S(s, s_b)] \tag{A63}$$

where $s_b$ is the size at birth and $t_b$ is the time at birth.

To map these to our inheritance matrix model, we observe that samples from *Equation A63* follow

$$s_{d,m} = \tilde{g}(t_{b,m}, s_{b,m}) + \tilde{\eta}_m(t_{b,m}, s_{b,m}) \tag{A64}$$

where $\tilde{g}(t_{b,m}, s_{b,m}) = E_P[s_d|t_b, s_b]$ is a drift term and $\tilde{\eta}_m$ is a zero-mean noise term that depends both on time of day and birth size. Note that both $\tilde{g}(t_{b,m})$ and $\tilde{\eta}_m(t_{b,m}, s_{b,m})$ are periodic functions of time at birth $t_{b,m}$. Since the latter is not explicitly modelled in our framework, here, we replace it with the state $x_{0,m}$ of the circadian clock, such that the update equations in *Equation A64* now appear as

$$s_{d,m} = g(x_{0,m}, s_{b,m}) + \eta_m(x_{0,m}, s_{b,m}). \tag{A65}$$

To gain intuition into the shape of the unknown functions $g$ and $h$, we linearise the equations around some basal level $x = \delta$ of a clock-less mutant, which gives

$$s_{d,m} = g(\delta, s_{b,m}) + g'(\delta, s_{b,m})x_0 + \eta_m(\delta, s_{b,m}) + x_0\eta_m'(\delta, s_{b,m}), \tag{A66}$$

For simplicity assume $\eta_m'(\delta, s_{b,m}) = 0$ and that the clock-less mutant follows a linear cell size control model with gamma-distributed size increments $\phi_{A,m} \sim$ Gamma with mean $\Delta$ as in *Martins et al., 2018*. These assumptions lead to the relations,

$$g(\delta, s_{b,m}) = \Delta + as_{b,m}, \quad \eta_m(\delta, s_{b,m}) = \phi_{A,m} - \Delta. \tag{A67}$$

Using $s_{b,2m} = s_{b,2m+1} = s_{d,m}/2$, we can obtain the linearised inheritance matrix model equations for the circadian cell size control model (**Appendix 1—figure 6e**):

$$s_{b,2m} = \tfrac{1}{2}(as_{b,m} + bx_{0,m} + \xi_m), \quad \xi_{2m} = \phi_{A,m},$$
$$s_{b,2m+1} = \tfrac{1}{2}(as_{b,m} + bx_{0,m} + \xi_m) \quad \xi_{2m+1} = \phi_{A,m}$$

(A68)

where now $\xi_m$ is the added size and $x_{0,m}$ is the output $x_{0,m} = x_{1,m} + x_{2,m}$ of a circadian oscillator governed by

$$\begin{pmatrix} x_{1,n+1} \\ x_{2,n+1} \end{pmatrix} = x_{0,n+1} = \hat{\theta}x_{0,n} + \phi_{n+1},$$

(A69)

for cell generation $n$, where $\phi = (\phi_1, \phi_2)^\top$ are noise terms added to the elements of $x_0$ and $\hat{\theta}$ is some complex eigenvalued $2 \times 2$ inheritance matrix given by

$$\hat{\theta} = \begin{pmatrix} D\cos\dfrac{2\pi}{P} & D\sin\dfrac{2\pi}{P} \\ -D\sin\dfrac{2\pi}{P} & D\cos\dfrac{2\pi}{P} \end{pmatrix}.$$

(A70)

Following this, we see that the circadian clock is incorporated into this cell size control system in the same way as the kicked cell cycle model outlined in the previous section (Appendix 1 - Section A7.1). Using **Equation A62** we can write the interdivision time of a cell with index $p$ as

$$\tau_p = \dfrac{\ln(\dfrac{as_{b,p} + bx_{0,p} + \xi_p}{s_{b,p}})}{\alpha}.$$

(A71)

Then taking the vector of cell cycle factors for the mother cell to be $y_m = (y_{m,1}, y_{m,2}, y_{m,3}, y_{m,4})^\top = (s_{b,m}, x_{1,m}, x_{2,m}, \xi_m)^\top$ and comparing **Equation 1a** and **b** with **Equation A68** and **Equation A71** we obtain

$$\begin{aligned} f(y) &= \dfrac{\ln(\frac{ay_1 + b(y_2 + y_3) + y_4}{y_1})}{\alpha} \\ g(y) &= (\tfrac{1}{2}(ay_1 + b(y_2 + y_3) + y_4), \; D\cos\dfrac{2\pi}{P}y_2 + D\sin\dfrac{2\pi}{P}y_3, \; -D\sin\dfrac{2\pi}{P}y_2 + D\cos\dfrac{2\pi}{P}y_3, \; cy_4)^\top \\ \beta &= (0, \; \mathrm{E}[\hat{\phi}_1], \; \mathrm{E}[\hat{\phi}_2], \; \mathrm{E}[\hat{\phi}_A])^\top. \end{aligned}$$

(A72)

Computing the means using **Equation A1** we get

$$\mu_1 = \dfrac{-\mu_\phi}{a-2}, \quad \mu_2 = 0, \quad \mu_3 = 0, \quad \mu_4 = \mu_\phi.$$

(A73)

Then using **Equation A72** in **Equation A5** and **Equation A12**, we can solve for the means

$$\alpha = \begin{pmatrix} 1 \\ 1 \\ 1 \\ 1 \end{pmatrix} \quad \text{and} \quad \theta = \begin{pmatrix} \dfrac{a}{2} & \tfrac{1}{2}(a-2) & \tfrac{1}{2}(a-2) & \tfrac{1}{2}(a-2) \\ 0 & D\cos(\dfrac{2\pi}{P}) & D\sin(\dfrac{2\pi}{P}) & 0 \\ 0 & -D\sin(\dfrac{2\pi}{P}) & D\cos(\dfrac{2\pi}{P}) & 0 \\ 0 & 0 & 0 & 0 \end{pmatrix}.$$

(A74)

Then taking $\tilde{S}_2 = 0$ and using **Equation A13**, we obtain the following for $S_1$:

$$S_1 = \dfrac{(a-2)^2}{4\alpha^2} \begin{pmatrix} 0 & 0 & 0 & 0 \\ 0 & b^2\eta_1^2 & b^2\mathrm{cor}_{12}\eta_1\eta_2 & b\mathrm{cor}_{A1}\eta_1\eta_2 \\ 0 & b^2\mathrm{cor}_{12}\eta_1\eta_2 & b^2\eta_2^2 & b\mathrm{cor}_{A2}\eta_2\eta_A \\ 0 & b\mathrm{cor}_{A1}\eta_1\eta_A & b\mathrm{cor}_{A2}\eta_2\eta_A & \eta_A^2 \end{pmatrix}$$

(A75)

where $\text{cor}_{ij}$ indicates the correlation between a pair of noise terms $\phi_i$ and $\phi_j$, and $\eta_i^2 = \frac{\text{Var}(\phi_i)}{\mu_\phi^2}$ for $i,j \in \{1, 2, A\}$.

### A7.3 Model comparison

We notice that the kicked cell cycle model has three cell cycle factors, while the circadian-clock-driven cell size control model has four cell cycle factors. The eigenvalues of the inheritance matrix $\boldsymbol{\theta}$ determining the correlation patterns are

$$\boldsymbol{\lambda} = (\beta, De^{+\frac{2\pi i}{P}}, De^{-\frac{2\pi i}{P}})^\top \tag{A76}$$

for the kicked cell cycle model and

$$\boldsymbol{\lambda} = (0, \frac{a}{2}, De^{+\frac{2\pi}{P}}, De^{-i\frac{2\pi}{P}})^\top \tag{A77}$$

and for the cell size control model. In both models, either the complex pair of eigenvalues $De^{\pm\frac{2\pi i}{P}}$ produces oscillatory behaviour. The overall correlation patterns are of mixed type, depending on the parameters $\beta$ and $a$.

To compare the models quantitatively, we match their mother-daughter interdivision time correlation coefficient in the absence of clock coupling. For the kicked cell cycle model, we notice that $\rho_{(1,0)} = \beta$ in the absence of clock coupling. The cell size control model reduces to the model in Appendix 1 - Section A6.1 in the absence of clock coupling, which satisfies $\rho_{(1,0)} = \frac{a-2}{4}$. Since realistic cell size control mechanisms (*Taheri-Araghi et al., 2015*; *Amir, 2014*; *Tanouchi et al., 2015*; *Sauls et al., 2016*) ($a \in [0, 2)$) ranging from sizers ($a = 0$) to adders ($a = 1$) to timers ($a = 2$) imply $\beta \leq 0$, we find that the kicked cell cycle obeys a mixed correlation pattern of the alternator/oscillator type while the cell size control model obeys a aperiodic/oscillator pattern.

Focusing on the common adder size control ($a = 1$), we find that the regions where the cousin-mother inequality is satisfied is remarkably similar in both models when $\beta$ is matched accordingly (*Appendix 1—figure 6b and f*). The lineage correlation function (red line) oscillates but the cross-branch correlation functions (blue line) alternates for the kicked cell cycle (*Appendix 1—figure 6c–d*) but not for the cell size control model (*Appendix 1—figure 6g–h*).

## A8. Inference validation using simulated data

To validate the inference results discussed in the main text we simulate interdivision time data using the maximum posterior parameters from the inference on two of the original live imaging datasets, and compare the output and model fit to our original inference.

We take the maximum posterior parameter sets from the original inference on two datasets (*Appendix 1—table 2*), cyanobacteria and mouse embryonic fibroblasts, and produce simulated interdivision time lineage data in *MATLAB* using custom scripts and Random Trees (*Kaj and Gaigalas, 2022*). We chose to look at these two datasets in order to analyse the posterior distribution of the inferred underlying period $T_{-1}$ to compare to the approximately 24 hr results seen in the main text.

**Appendix 1—table 1.** Lineage tree statistics obtained from each dataset used in this work.
Mean interdivision time, $\bar{\tau}$ tree variance, $\hat{s}_\tau$ CVs and all correlation coefficients ± standard deviation of the bootstrap distributions from 10,000 re-samplings with replacement. Statistics were calculated on all available cells that could be put in the required family pair (Materials and methods - 'Data analysis and Bayesian inference of the inheritance matrix model'). Shaded datasets exhibit the cousin-mother inequality.

| Cell type | Mean $\bar{\tau}$ (hours) | Variance $\hat{s}_\tau$ (hours²) | CV | $\hat{\rho}_{\text{md}}$ | $\hat{\rho}_{\text{gg}}$ | $\hat{\rho}_{\text{ss}}$ | $\hat{\rho}_{\text{cc}}$ | 1D AiC | 2D AiC | ref. |
|---|---|---|---|---|---|---|---|---|---|---|
| Cyanobacteria (*S. elongatus*) | 15.47±3.27 | 10.67±0.36 | 0.21±0.004 | −0.25±0.024 | −0.16±0.028 | 0.63±0.028 | 0.40±0.019 | 408.13 | 14.01 | *Martins et al., 2018* |
| Clock deleted cyanobacteria (*S. elongatus ΔkaiBC*) | 14.43±1.89 | 3.57±0.15 | 0.13±0.003 | −0.02±0.027 | 0.12±0.032 | 0.48±0.025 | 0.26±0.021 | 172.47 | 14.00 | *Martins et al., 2018* |
| Mycobacteria (*M. smegmatis*) | 2.52±0.65 | 0.42±0.03 | 0.26±0.010 | −0.16±0.041 | −0.05±0.051 | 0.55±0.033 | 0.05±0.040 | 8.69 | 14.01 | *Priestman et al., 2017* |

*Appendix 1—table 1 Continued on next page*

*Appendix 1—table 1 Continued*

| Cell type | Mean $\bar{\tau}$ (hours) | Variance $\hat{s}_{\tau}$ (hours²) | CV | $\hat{\rho}_{md}$ | $\hat{\rho}_{gg}$ | $\hat{\rho}_{ss}$ | $\hat{\rho}_{cc}$ | 1D AiC | 2D AiC | ref. |
|---|---|---|---|---|---|---|---|---|---|---|
| Human colorectal cancer (*HCT116*) | 16.39±2.55 | 6.49±1.10 | 0.15±0.012 | 0.07±0.141 | −0.08±0.227 | 0.73±0.047 | 0.34±0.070 | 22.20 | 14.23 | *Chakrabarti et al., 2018* |
| Neuroblastoma (*TET21N*) | 17.12±3.13 | 9.79±0.68 | 0.18±0.006 | 0.35±0.027 | 0.15±0.022 | 0.69±0.021 | 0.40±0.018 | 196.79 | 14.00 | *Kuchen et al., 2020* |
| Mouse embryonic fibroblasts (*NIH3T3*) | 20.40±6.09 | 37.03±4.31 | 0.30±0.015 | 0.39±0.040 | −0.01±0.057 | 0.59±0.029 | 0.22±0.047 | 21.64 | 14.01 | *Mura et al., 2019* |

**Appendix 1—table 2.** Maximum posterior matrices from the original inference, used to simulate interdivision time trees used for analysis in Appendix 1 - Section A8 *Appendix 1—figure 9*.

| Matrix | Cyanobacteria (S.elongatus) | Mouse embryonic fibroblasts (*NIH3T3*) |
|---|---|---|
| $\boldsymbol{\theta}$ | $\begin{pmatrix} -0.561848009 & -0.144058395 \\ 1.534655933 & -0.255834609 \end{pmatrix}$ | $\begin{pmatrix} -0.417019954 & -1.401854729 \\ 0.544365633 & 1.127838871 \end{pmatrix}$ |
| $\boldsymbol{S_1}$ | $\begin{pmatrix} 2.373007424 & 0.097863327 \\ 0.097863327 & 1.410419383 \end{pmatrix}$ | $\begin{pmatrix} 103.123125667 & -83.980021238 \\ -83.980021238 & 80.112064942 \end{pmatrix}$ |
| $\boldsymbol{S_2}$ | $\begin{pmatrix} 0 & 0 \\ 0 & 0 \end{pmatrix}$ | $\begin{pmatrix} 0 & 0 \\ 0 & 0 \end{pmatrix}$ |
| $\boldsymbol{\alpha}$ | $\begin{pmatrix} 1 \\ 1 \end{pmatrix}$ | $\begin{pmatrix} 1 \\ 1 \end{pmatrix}$ |

From this simulated data, the correlation coefficients are calculated using the methods outlined in Materials and methods - 'Data analysis and Bayesian inference of the inheritance matrix model', and then we look at the model inference on these new, simulated correlations, to compare to the original. These simulations produce correlation patterns that reproduce the experimentally measured correlations (comparing *Appendix 1—figure 9a–b* with *Figure 3a and f*).

The posterior distribution of the simulated patterns are the same for the cyanobacteria, exhibiting an 100% oscillator pattern (*Appendix 1—figure 9a*), matching the fitting to the original dataset (*Figure 3a*). Mouse embryonic fibroblasts (*Appendix 1—figure 9b*) loses some of it's original 100% oscillator pattern (*Figure 3f*) in favour of an alternator pattern. However, an oscillator pattern is still dominant.

We see that for cyanobacteria (*Appendix 1—figure 9c*) and mouse embryonic fibroblasts (*Appendix 1—figure 9d*), the posterior distribution for the inference on the simulated data for the correlation function oscillatory period, $T_{-1}$ (*Appendix 1—figure 9c and d*), exhibits a large overlap with the original posterior distribution discussed in Circadian oscillations in cyanobacteria and fibroblastssupport coupling of the circadian clock and the cell cycle (*Figure 5e*). The difference in the median for these posterior distributions is 0.42 hr for mouse embryonic fibroblasts (*Appendix 1—figure 9d*) and just 0.11 h for cyanobacteria (*Appendix 1—figure 9c*). This result validates our analysis of these posterior distributions showing that the period that we reconstruct from the simulated correlation patterns is consistent with the original data.

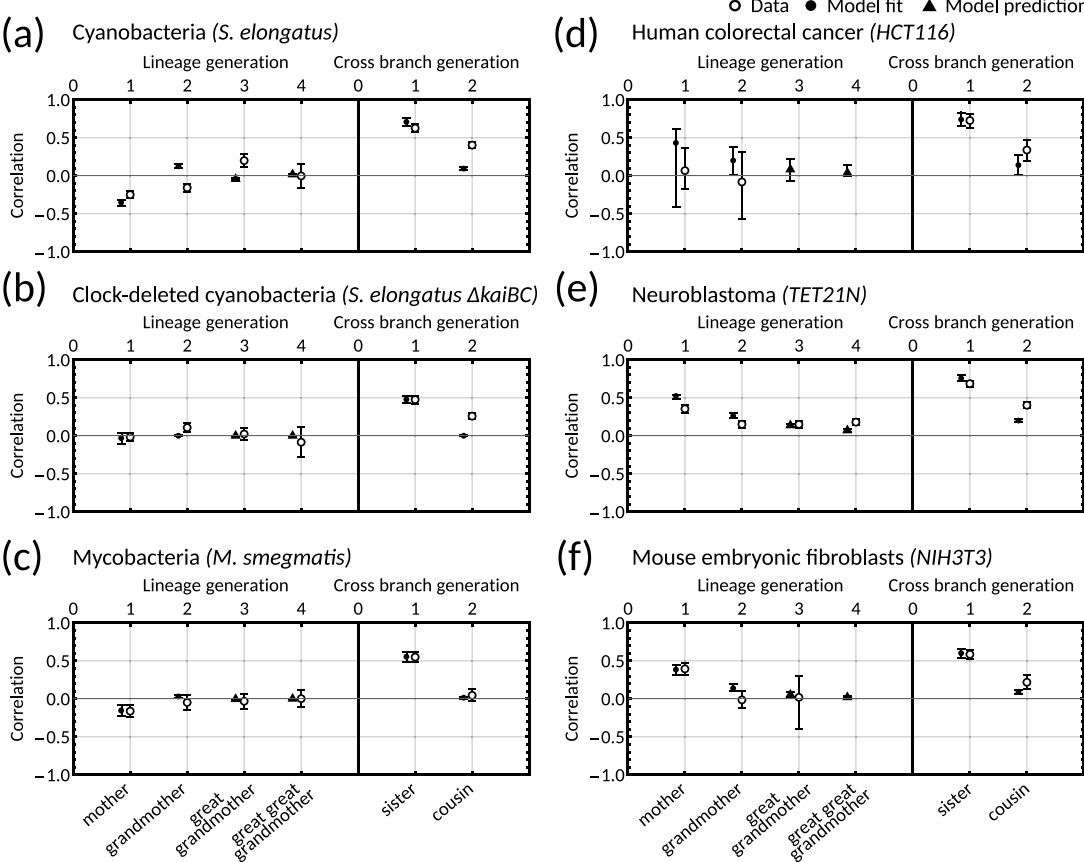

**Appendix 1—figure 1.** One-dimensional model with simple inheritance rules results in a poor fit for datasets displaying the cousin-mother inequality. (**a–f**) Plots showing data (open markers) against model predictions (solid black) for the one-dimensional model **Cowan and Staudte, 1986** for (**a**) cyanobacteria, (**b**) clock-deleted cyanobacteria, (**c**) mycobacteria, (**d**) human colorectal cancer, (**e**) neuroblastoma and (**f**) mouse embryonic fibroblasts. We fit the model using the same likelihood function (**Equation M10**) and methods (Materials and methods - 'Data analysis and Bayesian inference of the inheritance matrix model') as in the main text. Points (black) give the median model output for each correlation and error bars give the 95% bootstrapped confidence intervals from 10,000 re-samplings with replacement. Circular points show the model fitted correlations (mother-daughter, grandmother-granddaughter, sister-sister and cousin-cousin) whereas triangular points demonstrate model predictions. For this fitting we used 100,000 samples (in contrast to 10 million used in the main text).

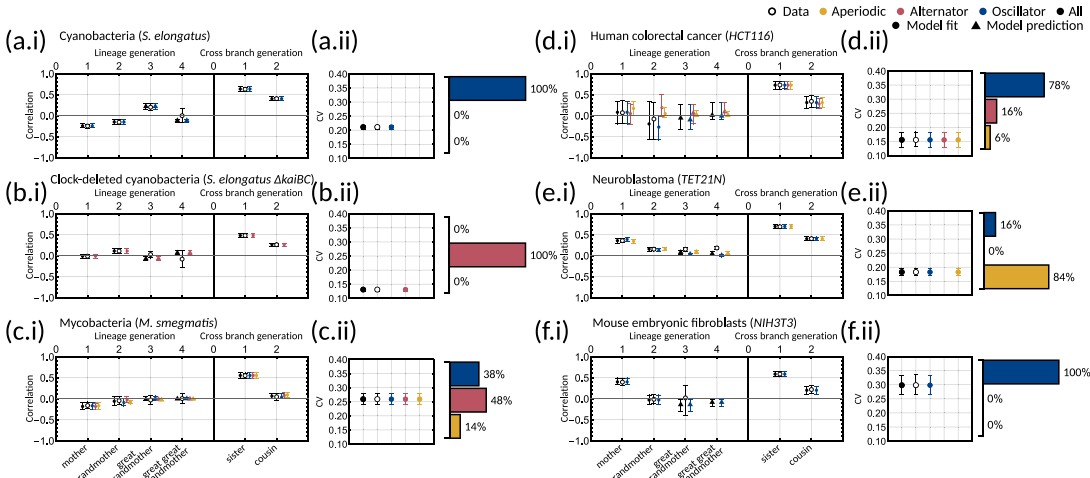

**Appendix 1—figure 2.** Bayesian inference demonstrates that multiple correlation patterns can explain the experimental data. (**a-f.i**) Plots of model fits and predictions (solid markers) against the data (open markers) for the family pair correlation coefficients for (**a**) cyanobacteria, (**b**) clock-deleted cyanobacteria, (**c**) mycobacteria, (**d**) human colorectal cancer, (**e**) neuroblastoma and (**f**) mouse embryonic fibroblasts. Colours of the solid markers represent the fits and predictions for parameter samples clustered by correlation pattern. Inset for each panel is a bar chart giving the distribution of the three patterns for each dataset. (**a-f.ii**) Plots of model output against the data for the interdivision time covariance. In this figure, the error bars for the data (unfilled black points) are calculated via bootstrapping of 10,000 samples with replacement to give the 95% confidence interval. For the model, error bars represent the 95% credible interval, computed by taking the 2.5th and 97.5th percentile of the sampled values. For all plots, circles indicate fitted correlations and triangles show predicted correlations. We can see that the model fit is good for all datasets as the error bars overlap with that of the data, and this is reflected in the low AIC given in *Appendix 1—table 1*.

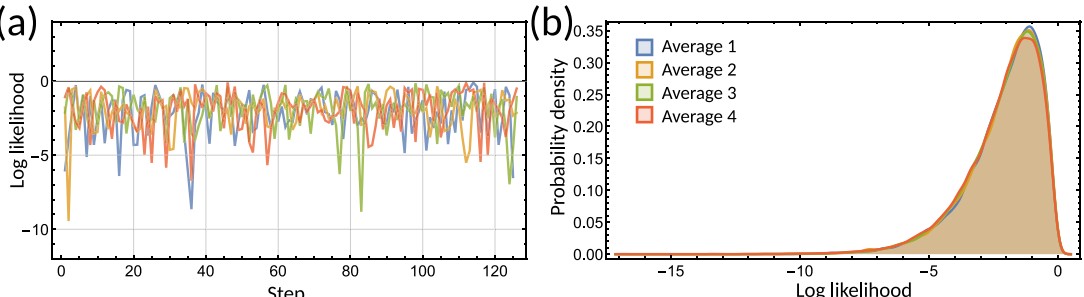

**Appendix 1—figure 3.** The log-likelihood converges during the parameter inference. (**a**) Trace of the log-likelihood from four initialisations of the inference on the clock-deleted cyanobacteria dataset (different colours). (**b**) Histogram of the posterior distribution of the log-likelihood for the inference samples on the clock-deleted cyanobacteria dataset. The histogram for each average aligns demonstrating convergence of the log-likelihood.

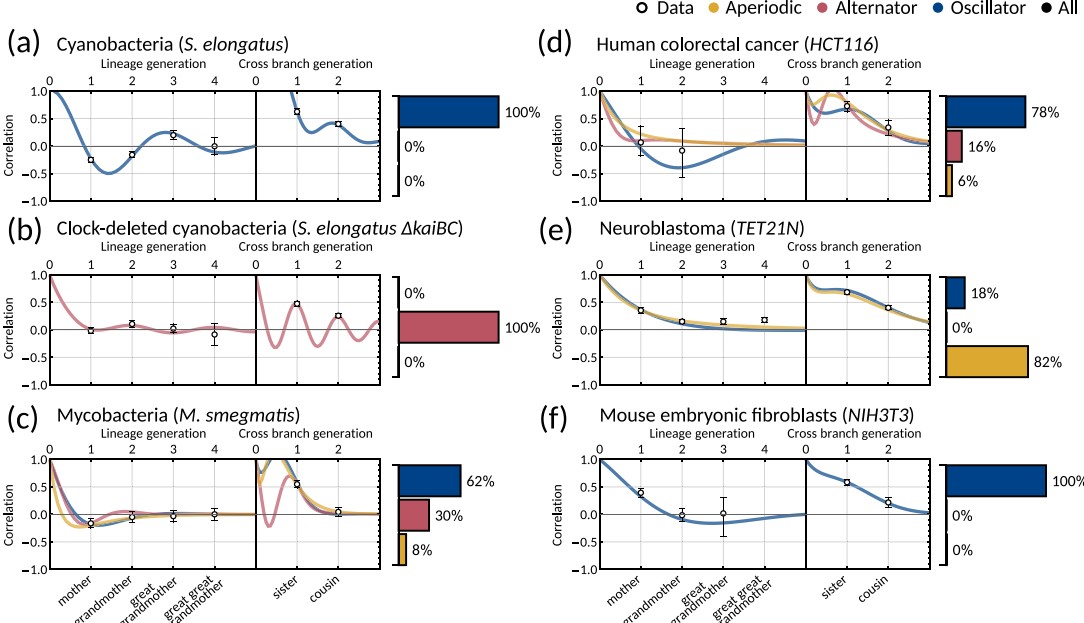

**Appendix 1—figure 4.** Two-dimensional inheritance matrix model gives a good fit for $\boldsymbol{\alpha} = (1, 0)^\top$. Same panels as in *Figure 3* but with $\boldsymbol{\alpha} = (1, 0)^\top$ and showing only one sample. We show the calculated family correlations with 95% bootstrapped confidence intervals (open markers) and a single sample of the model fit for (**a**) cyanobacteria, (**b**) clock-deleted cyanobacteria, (**c**) mycobacteria, (**d**) human colorectal cancer, (**e**) neuroblastoma and (**f**). Posterior parameter sets are clustered by correlation patterns (bar charts.) For this fitting we used 100,000 samples (in contrast to 10 million used in the main text). We see a similar fit and pattern distributions for all cell types except for mycobacteria (**c**), which here displays a dominant oscillator pattern.

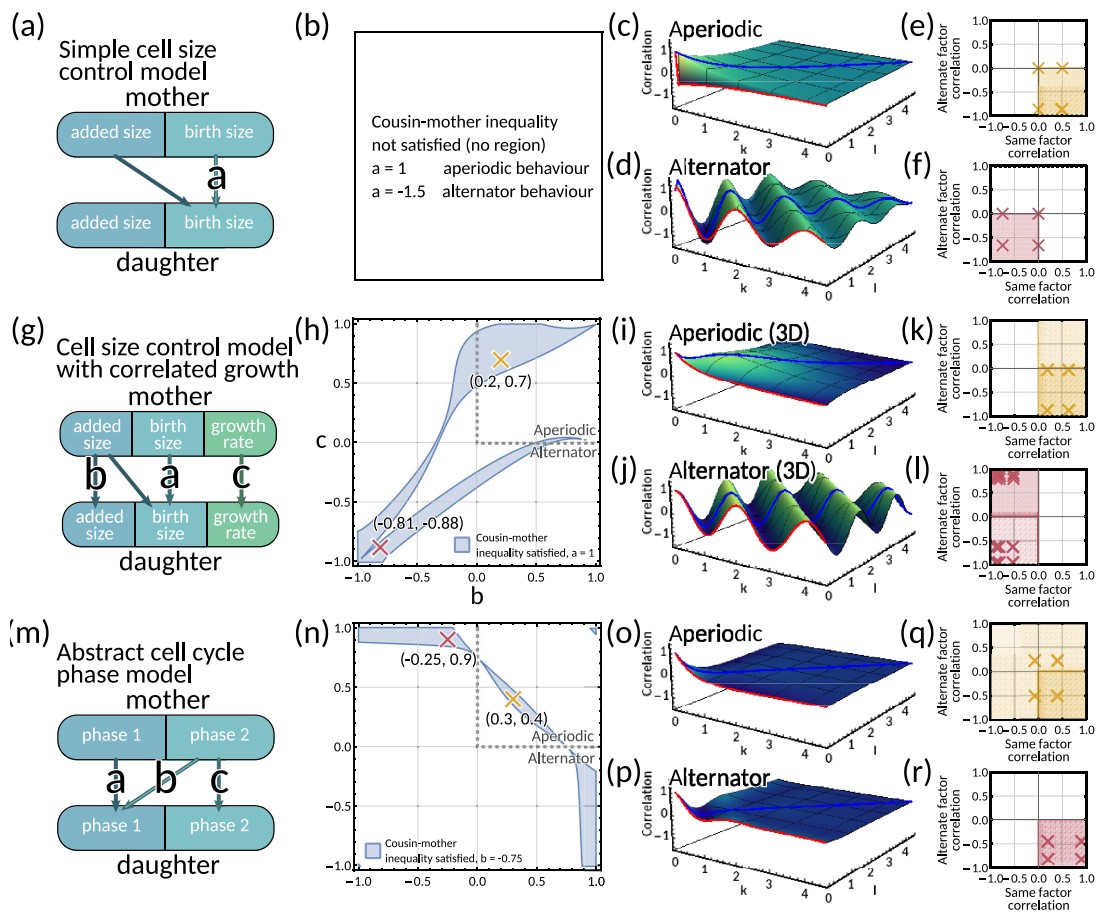

**Appendix 1—figure 5.** Mapping mechanistic models to the inheritance matrix model framework. (**a–f**) *Simple cell size control model.* (**a**) Model schematic. (**b**) The cousin-mother inequality cannot be satisfied for any choice of parameter $a$. (**c–d**) Generalised tree correlation function plots (**c**) for $a = 1$ and (**d**)$a = -1.5$ resulting in aperiodic and an alternator pattern respectively. (**e–f**) Same vs alternate factor mother-daughter correlation plots for (**e**)$a = 1$ and (**f**)$a = -1.5$. In panels (**b–f**) we fix $E[\xi] = 1, Var(\xi) = 0.1, \kappa = 1$. (**g–l**) *Cell size control model with correlated growth rate.* (**g**) Model schematic. (**h**) Region plot with fixed parameter $a = 1$ showing the parameter space $b, c$ in $(-1, 1)$ that satisfies the cousin-mother inequality (blue). Example parameter choices are also plotted for an aperiodic (yellow) and an alternator (red) pattern. (**i–j**) Generalised tree correlation function plots for (**i**)$(b, c) = (0.2, 0.7)$ and (**j**)$(b, c) = (-0.81, 0.88)$ resulting in aperiodic and an alternator pattern respectively. (**k,l**) Same vs alternate factor mother-daughter correlation plots for (**k**)$(b, c) = (0.2, 0.7)$ and (**l**)$(b, c) = (-0.81, 0.88)$. In panels (**h–l**) we fix $E[\xi] = E[\phi] = 1, Var(\xi) = Var(\phi) = 1, \kappa = 1$. (**m–r**) *Two cell cycle phase model* (**m**) Model schematic. (**n**) Region plot with fixed parameter $b = -0.75$ showing the parameter space $a, c$ in $(-1, 1)$ that satisfies the cousin-mother inequality (blue). Example parameter choices are also plotted for an aperiodic (yellow) and an alternator (red) pattern. (**o–p**) Generalised tree correlation function plots (**o**) for $(a, c) = (0.3, 0.4)$ and (**p**)$(a, c) = (-0.25, 0.9)$ resulting in aperiodic and an alternator pattern respectively. (**q–r**) Same vs alternate factor mother-daughter correlation plots for (**q**)$(a, c) = (0.3, 0.4)$ and (**r**)$(a, c) = (-0.25, 0.9)$. In panels (**n–r**) we fix $Var(\xi) = Var(\phi) = 1$.

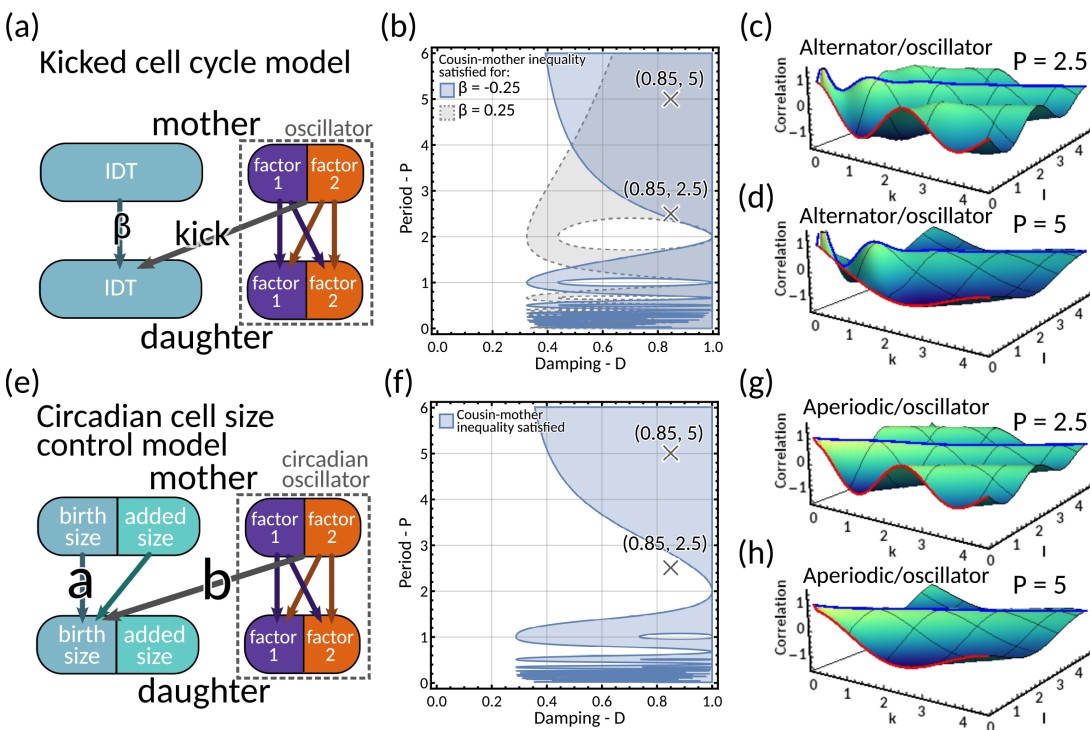

**Appendix 1—figure 6.** Models of circadian-clock-driven correlation patterns (**a–d**) *Kicked cell cycle model*. (**a**) Model schematic. The mother to daughter IDT inheritance is given by $\beta = \frac{(a-2)}{4}$ where $a$ is the size control parameter. The 'kick' to the cell cycle us produced by a two-dimensional complex eigenvalued inheritance matrix model system with oscillator behaviour. (**b**) Region plot for $\beta = -0.25$ (blue) and $\beta = 0.25$ (grey), demonstrating the region for this model where the cousin inequality is satisfied. Here we fix the variances of the noise terms $\xi_1, \xi_2$ and $\xi_\tau$ all equal to 0.1. (**c–d**) Plot of the generalised tree correlation function for (**c**) $(D, P) = (0.85, 2.5)$ and (**d**) $(D, P) = (0.85, 5)$. In both these plots we take $\beta = -0.25$, meaning the model has a mixture of alternator and oscillator behaviours. The cousin inequality is satisfied for both these parameter choices. (**e–h**) *Circadian cell size control model* (**e**) Model schematic. The parameter $a$ gives how the daughter's birth size depends on the mother's birth size; and $b$ gives the coupling of the circadian oscillator to the size control. (**f**) Region plot demonstrating where the cousin inequality is satisfied. We fix $a = 1, b = 1$. Correlations between noise terms are fixed equal to 0 and we set $\eta_i = 0.1$ for $i \in \{1, 2, A\}$. (**g–h**) Plots of the generalised tree correlation function for the same fixed parameters specified in panel (**f**), with (**g**) $(D, P) = (0.85, 2.5)$, and (**h**) $(D, P) = (0.85, 5)$. As we fix $a = 1$, these plots show a combination of aperiodic and oscillator behaviour. We note that for $(D, P) = (0.85, 2.5)$, the cousin inequality is not satisfied. This demonstrate that oscillatory behaviour is not a necessary condition for the cousin inequality to be satisfied.

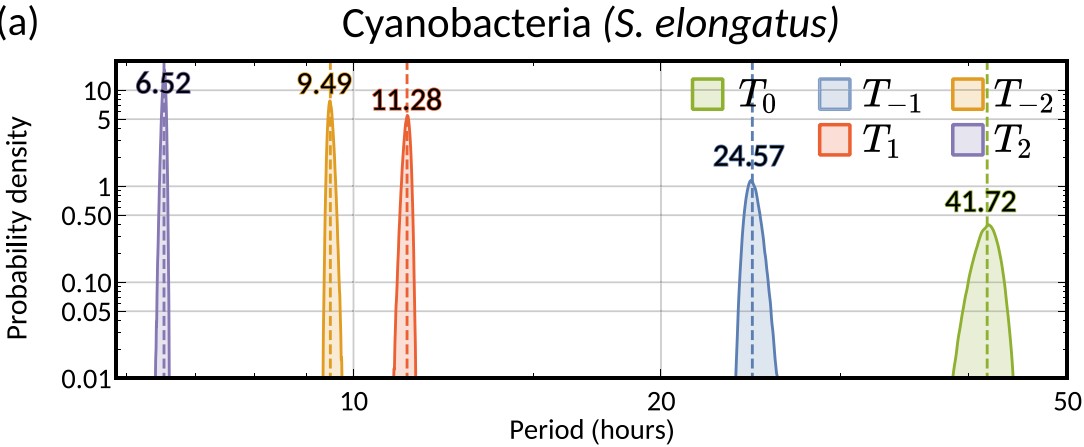

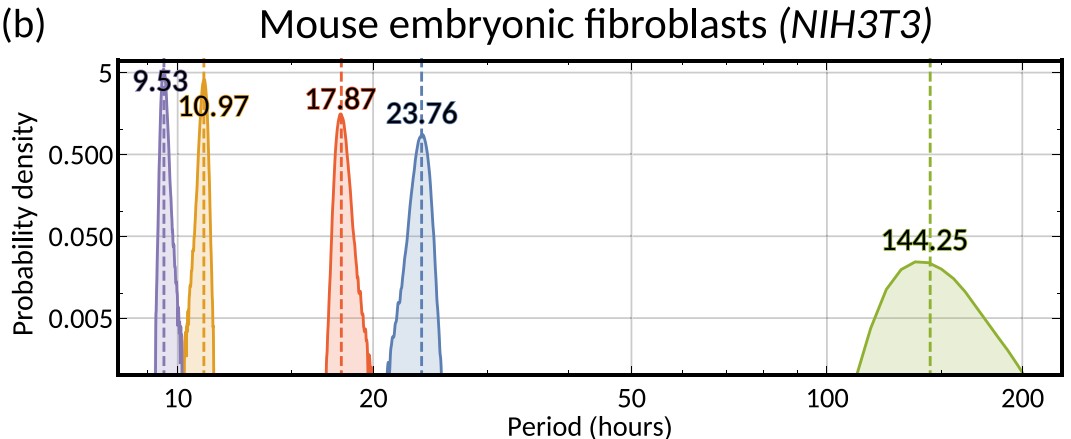

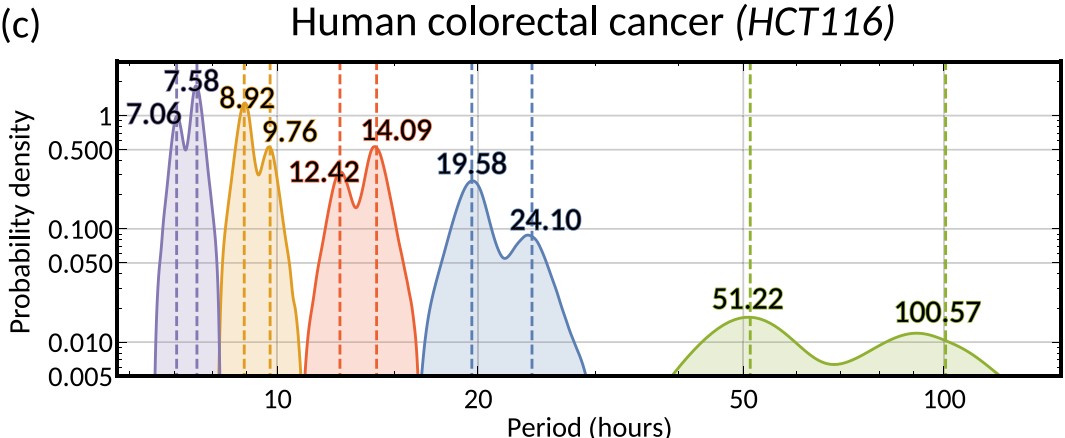

**Appendix 1—figure 7.** A range of oscillator periods can explain oscillatory interdivision time patterns. Histogram of the posteriors of the possible periods underlying the lineage correlation function for (**a**) cyanobacteria, (**b**) mouse embryonic fibroblasts and (**c**) human colorectal cancer, calculated using *Equation 7*. Numerical values give medians of the posterior distributions for each $T_n$. For (**c**) human colorectal cancer, we take the median period of each cluster where the clusters are allocated through the sign of the real part of the eigenvalue (see *Figure 5f*). For all panels the correlation oscillation period $T_0$ is given in green and the oscillator periods in different colours. The period analysed in 'The inheritance matrix model predicts the hidden dynamical correlations of cell cycle factors' corresponds to the histograms of $T_{-1}$ (blue).

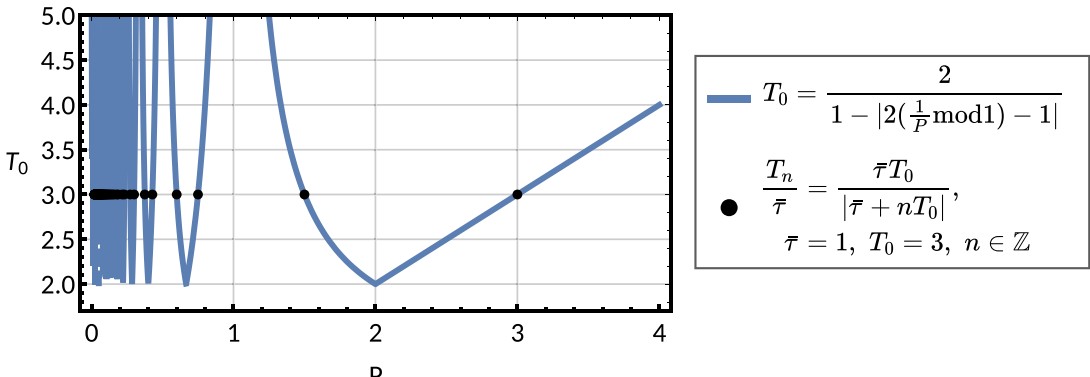

**Appendix 1—figure 8.** Observed period $T_0$ against chosen period parameter $P$ for a forced oscillator pattern. Plot of the function for $P$ against the observed lineage correlation function period $T_0$ given in **Equation M9** (blue line), for an oscillator pattern given in 'The inheritance matrix model reveals three distinct interdivision time correlation patterns'. We see that $T_0 = P$ for $P > 2$. For chosen $T_0 = 3$ with $\tau = 1$ and various $n$ we see how the parameters $P$ that produce the corresponding $T_0$s are directly equal to the possible $T_n$ we can derive from the chosen $T_0$ (black points), using **Equation 7**.

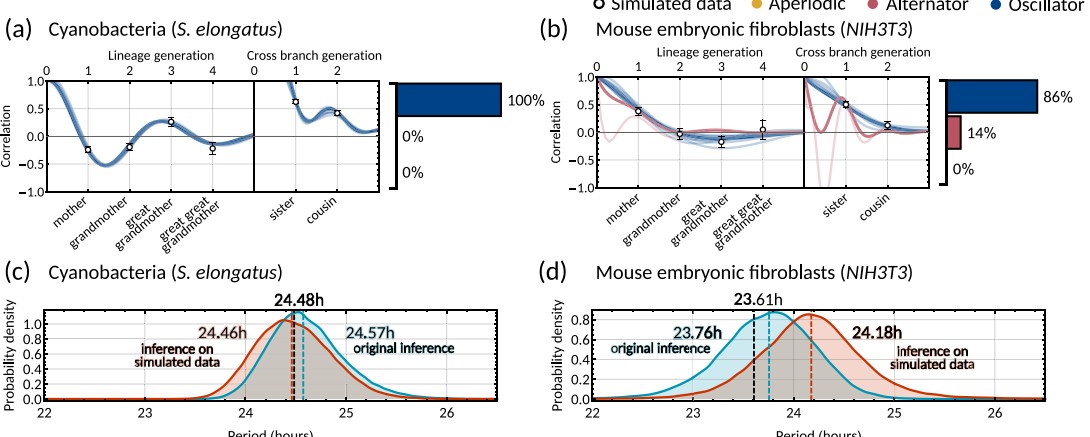

**Appendix 1—figure 9.** Validation of the Bayesian inference method using simulated data. Model fits and distribution of patterns for data simulated using the maximum posterior parameter set (**Appendix 1—table 2**) for (**a**) cyanobacteria, and (**b**) mouse embryonic fibroblasts. To simulate interdivision time lineage trees, we take the maximum posterior parameter sets from the original inference on the two datasets. These trees are simulated using **Equation 2a** in MATLAB using custom scripts which utilise 'Random trees' branching process (**Kaj and Gaigalas, 2022**). For each dataset, we first simulate a complete tree of 11 generations (2047 cells) and take the last 1000 cells to sample stationary initial conditions. For the final simulated data, we simulated a number of smaller trees of 6 generations (63 cells each) to better represent live imaging experiments. We divide the number of cells in the original dataset by 63 and simulate this number of trees, with each tree having initial condition sampled from the last 1000 cells of the original large tree. We then randomly sample 85% of the simulated cells without replacement to imitate loss of cells from imaging mid experiment. The calculation of the family interdivision time correlation coefficients and the parameter inference was done in the same way as with the original datasets as outlined in Materials and methods - 'Data analysis and Bayesian inference of theinheritance matrix model'. Pearson correlation coefficients (white dots) and 95% bootstrapped confidence intervals (error bars) were obtained through re-sampling with replacement (10,000 samples) of the simulated data. Posterior samples were clustered into aperiodic, alternator, and oscillator patterns (bar charts). We show several representative samples (solid and shaded lines) of the model fit drawn from the posterior distribution. We assume $\boldsymbol{\alpha} = (1, 1)^{\top}$. (**c–d**) Histograms of the inferred oscillator period $T_{-1}$ for the original inference (blue) and inference on the simulated data (orange) for cyanobacteria (**c**) and mouse embryonic fibroblasts (**d**), demonstrating significant overlap of the oscillator period of the simulated parameter set (black dashed line) and the posterior distribution from Bayesian inference. Note that the posterior distributions of the real (red) and simulated datasets (blue) also overlap. Dashed lines give the median period of these posterior distributions for original inference (blue) and inference on simulated data (orange). Maximum posterior parameters used in the simulations are given in **Appendix 1—table 2**.

**Appendix 1—table 3.** Comparison of different variance estimators.

Mean and 95% confidence intervals calculated from bootstrap distributions of 10,000 re-samplings with replacement for each dataset used in this work. The estimators are obtained as follows: bare variance is computed using all available cells that could be put in the required family pair (Materials and methods - 'Data analysis and Bayesian inference of theinheritance matrix model'). The lineage variance is calculated through the weighted variance with weights $w_i = 2^{-D_i}/N_{\text{trees}}$ following arguments similar to *Priestman et al., 2017*; *Nozoe et al., 2017*. Here $D_i$ is the number of divisions in the lineage that came before cell $i$ and $N_{\text{trees}}$ is the total number of trees in the whole dataset. The censored variance is calculated after pruning trees such that each tree contains lineages of the same length as in *Kuchen et al., 2020*; *Sandler et al., 2015*.

| Cell type | Bare variance (hours²) | Lineage variance (hours²) | Censored variance (hours²) |
|---|---|---|---|
| Cyanobacteria (*S. elongatus*) | 10.674 [9.966, 11.396] | 11.543 [10.420, 12.776] | 10.612 [9.850, 11.391] |
| Clock deleted cyanobacteria (*S. elongatus ΔkaiBC*) | 3.573 [3.288, 3.865] | 4.015 [3.529, 4.512] | 3.485 [3.176, 3.805] |
| Mycobacteria (*M. smegmatis*) | 0.427 [0.366, 0.494] | 0.601 [0.490, 0.716] | 0.609 [0.492, 0.738] |
| Human colorectal cancer (*HCT116*) | 6.489 [4.540, 8.809] | 7.357 [4.898, 10.262] | 6.741 [4.695, 9.124] |
| Neuroblastoma (*TET21N*) | 9.794 [8.539, 11.213] | 13.986 [10.735, 17.775] | 10.502 [8.554, 12.621] |
| Mouse embryonic fibroblasts (*NIH3T3*) | 37.032 [29.260, 46.162] | 46.378 [34.494, 60.090] | 39.418 [29.947, 50.219] |

