## [Editor Report]

This work makes an important contribution to the study of the cell cycle and inferring mechanisms by studying correlations in division timing between single cells. By treating the problem in a general way and computing over lineage trees, the authors can infer timescales in the underlying mechanism. The method is validated on data sets from bacterial and mammalian cells and can suggest when additional measurements are needed to distinguish competing models.

---

## [Decision Letter]

**Decision letter after peer review:**

Thank you for submitting your article "Patterns of interdivision time correlations reveal hidden cell cycle factors" for consideration by *eLife*. Your article has been reviewed by 2 peer reviewers, and the evaluation has been overseen by a Reviewing Editor and Aleksandra Walczak as the Senior Editor. The following individuals involved in the review of your submission have agreed to reveal their identity: Michael J Rust (Reviewer #1); Farshid Jafarpour (Reviewer #2).

Essential revisions:

1) As suggested by reviewer 1, please include mechanistic details for circadian rhythm models (e.g., showing the inheritance matrix). Also, explain why whether leading order approximations are valid for oscillatory models as suggested by the same reviewer.

2) As suggested by reviewer 2, more intuition is needed for some of the results. E.g., intuition why only eigenvalues of the inheritance matrix matter + intuition for the alternator pattern.

*Reviewer #1 (Recommendations for the authors):*

Concretely, while section 4 of the supplement shows how some simple mechanistic models map onto this formalism, none of these include an explicit circadian rhythm. Since the authors did develop a model like this in Martins et al., it would be valuable to show what the inheritance matrix, etc. are for this kind of model, and indeed whether going to leading order in fluctuations works satisfactorily.

*Reviewer #2 (Recommendations for the authors):*

The paper is well-written, scientifically sound, and easy to follow. I only have a few minor suggestions/comments:

1) It is not clear what the variance in Equation 4 is. Is it the variance of generation times across a single lineage or across the whole tree or the population variance? If I understand it correctly from the derivation in SI, it should be a lineage variance, which may not be immediately obvious given that rho is a tree variable. So it would be nice to specify which one it is.

2) I find it very surprising that only the eigenvalues of the inheritance matrix determine the correlation patterns, while all the other parameters, including how noises are correlated (anticorrelated) and whether growth factors have a positive or negative effect on the inter-division times, are irrelevant. It would be nice if the authors could provide an intuition for why that is the case.

3) The authors have used the AIC method for the goodness of fit. Initially, without being familiar with the method, I found it surprising that in the case of mycobacteria, a more general model could have a worse fit than its special case. I think a one-sentence explanation of this method could prevent such potential confusion for readers not familiar with this method.

4) What is the intuition behind the alternator pattern, i.e. why is there a period 2 oscillation in a model with real eigenvalues?

5) In the model, the inter-division time is a deterministic function of the cell-cycle factors (Equations 1 and 3a). The noise is only allowed in the inheritance of these factors, not in the division itself. This puts a constraint on what variables of a given model can be used as cell-cycle factors. I think this would be worth mentioning in the paper.

---

## [Author Response]

Essential revisions:1) As suggested by reviewer 1, please include mechanistic details for circadian rhythm models (e.g., showing the inheritance matrix). Also, explain why whether leading order approximations are valid for oscillatory models as suggested by the same reviewer.2) As suggested by reviewer 2, more intuition is needed for some of the results. E.g., intuition why only eigenvalues of the inheritance matrix matter + intuition for the alternator pattern.

We have addressed all reviewers' comments, which improved the manuscript significantly. We made the following changes:

a) We included a new section in the Appendix entitled “A7 Models of circadian-clock-driven correlation patterns”, where we derive the inheritance matrix of the kicked cell cycle model and the model of Martins et al. 2018 side-by-side. We also produced the new Appendix 1 – Figure A6 to compare the resulting correlation patterns.

b) We include a new section in the Appendix entitled “A2 Beyond the small noise approximation: cell cycle factor complexes account for nonlinear fluctuations”, where we derive the effect of nonlinearity on the resulting correlation patterns analytically. Such features can be described through extra cell cycle factors called complexes. However, additional complexes were not necessary to fit the present data.

c) We present a new formula, Equation 4, for the lineage tree correlation function highlighting the intuition of how the eigenvalues determine the correlation patterns while other factors determine their relative weights. We also included a new section in the Appendix entitled “A5 Solution of the tree correlation function and parameter identifiability for simple inheritance rules”, where we explain intuitively why the interdivision time correlation pattern carries no information on whether the growth factor increases or decreases growth.

d) We now clarify the meaning of our variance estimators and show that different estimators produce similar results. This is summarised in Appendix – Table A3.

Reviewer #1 (Recommendations for the authors):Concretely, while section 4 of the supplement shows how some simple mechanistic models map onto this formalism, none of these include an explicit circadian rhythm. Since the authors did develop a model like this in Martins et al., it would be valuable to show what the inheritance matrix, etc. are for this kind of model, and indeed whether going to leading order in fluctuations works satisfactorily.

Thank you for this inspiring comment. We have included new detailed analyses of modelling circadian clock-driven correlation patterns in Appendix 1 – Section A7. We include and compare both the kicked cell cycle model and the model from the Martins et al. 2018 PNAS paper; the latter models the circadian modulation of division rate and cell size control. We have derived the model's inheritance matrix and characterised its eigenvalues regarding cell size control parameters and the coupling to the circadian clock.

Within the IMM framework, the kicked cell cycle model and the Martins et al. 2018 model reduce to inheritance matrice with three and four cell cycle factors, respectively. The two models produce different correlation patterns corresponding to the oscillator and oscillator-alternator mixed patterns, respectively. Nevertheless, we find that only the two circadian factors are necessary to fit the experimental data with oscillatory patterns.

We understand that the division-coupling function described in Martins et al. 2018 is sharply peaked, while oscillations of cell cycle factors produced in the IMM are sinusoidal. Going beyond leading order fluctuations could account for such nonlinear oscillations. We now show that such nonlinearity can be modelled through extra cell cycle factors akin to cell cycle factor complexes. In our inference, however, we found that two cell cycle factors were sufficient to fit all circadian datasets. We here applied Occam's razor through Akaike’s Information Criterion, preferring a simple model over a more complex one when both fit the data (see also reply to comment 3 of Reviewer 2). Thus the circadian component of these oscillations is enough to characterise the interdivision time data.

Of course, the model by Martins et al. 2018 is more detailed as it includes cell size information. Including extra cell cycle factors, such as cell size, could significantly improve identifying the mechanisms generating these correlation patterns, but this is beyond the scope of our manuscript.

We realise there are some limitations to the linear model, and including nonlinear interactions via complexes could improve model identification. We now include the following paragraph in the discussion (lines 759-773):

“In principle, increasing the number of interacting cell cycle factors can lead to more complex composite patterns that involve combinations of the three patterns discussed in this paper, such as the alternator-oscillator (Appendix 1 – Figure A6c, d), aperiodic-oscillator (Appendix 1 – FigureA6g,h), or birhythmic correlation patterns. Such composite patterns could also arise as the result of nonlinear fluctuations that, within our framework, can be described by adding complexes of cell cycle factors to the inheritance matrix model (Appendix 1 – Section A2). The presence of such complexes induces higher-order harmonics in the correlation oscillations, similar to those observed in the cyanobacterial and mammalian circadian clock [12, 63], and detecting such complexes could provide an alternative route to increase the sensitivity of our inference method.”

Reviewer #2 (Recommendations for the authors):The paper is well-written, scientifically sound, and easy to follow. I only have a few minor suggestions/comments:1) It is not clear what the variance in Equation 4 is. Is it the variance of generation times across a single lineage or across the whole tree or the population variance? If I understand it correctly from the derivation in SI, it should be a lineage variance, which may not be immediately obvious given that rho is a tree variable. So it would be nice to specify which one it is.

Thank you for pointing this out. The variance derived in the Appendix, now Equation M2 of the main text, is indeed the lineage variance that is also equal to the variance from a tree where all lineages have the same number of generations. However, in the data analysis, we compute the variance over all cells used to compute the correlation coefficients, which equals the variance across the whole tree. We are aware that this approximation can introduce bias, and we made several changes to address this:

a) We now clarify in Section Methods D how the variance is computed from the data (line 872):

“Note that ŝ_τ_ is computed across all interdivision times used to calculate the correlation coefficients in each dataset.”

b) We now state clearly in Section Methods D the meaning of the variance in Equation M2 (line894):

“Note that (M2) is the interdivision time variance from a tree where all lineages have the same number of generations, which approximates the variance across all cells in the observed trees.”

c) We found that the various bias introduced was negligible for the data analysed. To quantify the variance bias introduced, we have added Appendix 1 – Table A3 that compares the tree variance statistics used in our inference to i) the variance estimator obtained from trees with equal generations per lineage, thus ignoring parts of the data similar to references [24] and [27] ; and ii) the variance computed from forward-lineage-weighting of the tree data, similar to what is done in references [22] and [66] of the main text. This suggests that the three estimators cannot be distinguished within the 95% confidence intervals for all six datasets.

d) We understand that this approximation has limitations. We now clarify how potential biases would affect our parameter inference and the identified correlation patterns. We found that a potential bias in the variance could lead to a biased estimate of the noise matrices *S*1,2 since multiplying the latter by a constant increases the variance by the same amount. The eigenvalues and the identified correlation patterns remain unaffected, as well as the conclusions of our work. Of course, our theory cannot predict this constant and an extended theory would be required to quantify such biases. This is beyond the scope of this manuscript, but we hope to address this in future work.

In summary, the identified correlation patterns remain unaffected, as well as the conclusions of our work. We have added Appendix 1 – Table A3:

2) I find it very surprising that only the eigenvalues of the inheritance matrix determine the correlation patterns, while all the other parameters, including how noises are correlated (anticorrelated) and whether growth factors have a positive or negative effect on the inter-division times, are irrelevant. It would be nice if the authors could provide an intuition for why that is the case.

Thank you for this comment. To provide an intuition as to how the correlation patterns depend on parameters, we have added a new formula that shows the lineage correlation function to be a weighted sum of powers of eigenvalues. This shows that the behaviour of the tree correlation as we change *k, l* corresponding to a pattern depends only on the eigenvalues, but the noise matrices ^S^1,2 modulate the weights, i.e., they can amplify or attenuate a correlation pattern.

This is now clarified in Results B.

It is an interesting question whether the correlation patterns of growth factors that have positive effects on the interdivision times differ from those that include factors with negative effects. we have included an entire section in the Appendix addressing this issue (Appendix Section A5). There we provide an analytical solution of the effect of a growth factor on the interdivision time. The correlation patterns indeed coincide for both growth-enhancing and attenuating cell cycle factors. The intuition behind this effect is that the pattern only measures the noise. Thus we cannot tell apart the effect of an attenuating factor, from a growth-enhancing factor.

Mathematically, we rescale all cell cycle factors such that each has a positive contribution to the noise pattern. Since we only deal with the rescaled cell cycle factors, we avoided this kind of unidentifiability. Of course, the issue only arises if the cell cycle factor was hidden and simultaneous measurement of cell cycle factor correlations could resolve whether such factors are growth-enhancing and attenuating.

3) The authors have used the AIC method for the goodness of fit. Initially, without being familiar with the method, I found it surprising that in the case of mycobacteria, a more general model could have a worse fit than its special case. I think a one-sentence explanation of this method could prevent such potential confusion for readers not familiar with this method.

To clarify the use of AIC we have added the following text to Results section D where we compare models for the 6 datasets (lines 349-360):

“We quantified the quality of our fits using the Akaike information criterion (AIC) (Methods D, (D2)) for each dataset and compared these to the one-dimensional model (Appendix 1 – Table A1). The AIC estimates the goodness of fit with a penalty for model complexity allowing us to select the simplest model that explains the data. The AIC values indicate that the inheritance matrix model with two cell cycle factors provides the simplest fit for all cell types used here, except for the mycobacteria data where simple inheritance rules provided an equally good fit with a significant reduction in the number of model parameters.”

Additionally we have added the following to the end of Methods D which gives the formula for the AIC.

During revisions we noted that the AIC values were erroneously stated for k=2 while indeed we used k=3 due to *S*_2_ ≠ 0 taking into account the sister-sister correlations. As a result, the 1D AIC reported in Table S1 increased slightly (by a value of 2). However this does not change the model selection or any other conclusion.

4) What is the intuition behind the alternator pattern, i.e. why is there a period 2 oscillation in a model with real eigenvalues?

We added some explanations to provide more intuition for the alternator pattern. The corresponding text reads now (lines 250-257):

“.. the alternator pattern generates oscillations with a fixed period of two generations in the lineage correlation function. The behaviour is typically observed for cell cycle factors with negative mother-daughter correlations (Appendix 1 – Section S6 a). In this case, we have at least one negative eigenvalue and thus (4) will alternate between positive and negative values for successive generations, producing the period two oscillation.”

We hope that our reply to point (2) helps to clarify this further.

(5) In the model, the inter-division time is a deterministic function of the cell-cycle factors (Equation 1 and 3a). The noise is only allowed in the inheritance of these factors, not in the division itself. This puts a constraint on what variables of a given model can be used as cell-cycle factors. I think this would be worth mentioning in the paper.

Noise added on ‘after’ division would be equivalent to adding an additional cell cycle factor *y*_i_ with corresponding elements of the θ matrix θ_*ij*_ = θ*_ji_* = 0 where *j* is the index of the other factors. Therefore this additional noise is already encapsulated in the IMM framework.

In the main text, Results A, we have added the following sentence to clarify this (lines 148-151):

“Note that we choose (1a) to be deterministic since division noise can be modelled by adding one more cell cycle factor that does not affect inheritance dynamics **g**.”